# BENCHMARKING THE FIDELITY AND UTILITY OF SYNTHETIC RELATIONAL DATA

## ABSTRACT

Synthesizing relational data has started to receive more attention from researchers, practitioners, and industry. The task is more difficult than synthesizing a single-table due to the added complexity of relationships between tables. For the same reason, benchmarking methods for synthesizing relational data introduces new challenges. Our work is motivated by a lack of an empirical evaluation of state-of-the-art methods and by gaps in the understanding of how such an evaluation should be done. We review related work on relational data synthesis, common benchmarking datasets, and approaches to measuring the fidelity and utility of synthetic data. We combine the best practices and a novel robust detection approach into a benchmarking tool and use it to compare six methods, including two commercial tools. While some methods are better than others, no method is able to synthesize a dataset that is indistinguishable from original data. For utility, we typically observe moderate correlation between real and synthetic data for both model predictive performance and feature importance.

## 1 INTRODUCTION

Synthesizing relational data - generating relational data that preserve the characteristics of the original data - is an emerging field. It promises several benefits, from protecting privacy to addressing data scarcity while preserving the complexities and inter-dependencies present in the original data. This makes it attractive for domains such as healthcare (Appenzeller et al., 2022), finance (Assefa et al., 2020), and education (Bonnéry et al., 2019), where accessing and utilizing data can be challenging due to privacy concerns, data scarcity, or biases (Ntoutsi et al., 2020; Rajpurkar et al., 2022).

The foundations of synthesizing relational data were laid by the Synthetic Data Vault (Patki et al., 2016). Recently several deep learning methods have been proposed (Gueye et al., 2023; Li & Tay, 2023; Mami et al., 2022; Xu et al., 2023; Canale et al., 2022; Solatorio & Dupriez, 2023; Pang et al., 2024). The field has also received attention from industry, with several commercial tools now available and with Google, Amazon, and Microsoft integrating them into their cloud services (Gretel.ai, 2024).

While there are several packages for evaluating the quality of synthetic data, only the SDMetrics package (Patki et al., 2016) provides some support for the evaluation of synthetic relational data. As such, the field lacks not only an empirical comparison of available methods but also an understanding of how such an evaluation should be done. We address this gap with an evaluation methodology that combines established evaluation metrics (Section 2.2), best practices, sampling procedures for relational data (Section 2.3), and a novel metric that improves on existing approaches and generalizes to relational data (Section 3). We implement the methodology in a benchmarking tool that is available as an open source package and can be easily extended. Finally, we use the benchmark for current state-of-the-art methods (Section 2.1) on several relational data sets (Section 2.3). This is the first comprehensive evaluation and comparison of methods for synthesizing relational data and provides valuable insights into their ability to synthesize relational aspects of the data (Section 4).

Our benchmark reveals that lenient evaluation practices in related work have led to sub-par fidelity in single-table generation. Furthermore, where individual tables are synthesized well, current methods struggle to faithfully model the relationships between them. We highlight these gaps and offer a robust evaluation tool to guide and assess future advancements in synthetic relational data.

## 2 RELATED WORK

### 2.1 METHODS FOR SYNTHESIZING RELATIONAL DATA

In this work we focus on relational data - a collection of tables connected by foreign keys that form a relational database. We distinguish this from synthesizing tabular data (a single-table), which is a special case and an even more active field (Borisov et al., 2022; Qian et al., 2023b). Here we briefly summarize the methods. A detailed description can be found in Appendix A.

The Synthetic Data Vault (SDV) uses Gaussian copulas and predefined distributions to model relational data. Row Conditional-TGAN (RC-TGAN) (Gueye et al., 2023) and Incremental Relational Generator (IRG) (Li & Tay, 2023) are based on GANs. The Realistic Relational and Tabular Transformer (REaLTabFormer) (Solatorio & Dupriez, 2023) and Composite Generative Models (Canale et al., 2022) are based on transformers. The work of Mami et al. (2022) is based on Graph Variational Autoencoders, while Xu et al. (2023) propose a framework for synthesizing many-to-many datasets using random graphs. Recently, Pang et al. (2024) propose ClavaDDPM, a method based on classifier-guided diffusion models.

### 2.2 METRICS FOR EVALUATING SYNTHETIC DATA

The two main aspects for evaluating the quality of synthetic tabular and relational data are *fidelity* and *utility*. Fidelity measures the degree of similarity between synthetic and real data in terms of its properties, whereas utility measures how well the synthetic data can replace real data when the data are part of some tasks, for example, for predictive modeling (Hansen et al., 2023). We further divide fidelity metrics into *statistical*, *distance-based*, and *detection-based* metrics. Utility of synthetic data is typically assessed with train-on-synthetic evaluate-on-real methods (Beaulieu-Jones et al., 2019).

Another dimension of evaluation metrics for relational data is granularity. The most common are *single-column* metrics that evaluate the marginal distributions, *two-column* metrics that evaluate bivariate distributions, *single-table* metrics that evaluate tables, and *multi-table* metrics that evaluate the relational aspects.

#### 2.2.1 STATISTICAL FIDELITY

Statistical fidelity methods are typically used to assess marginal distributions, sometimes bivariate distributions. The most commonly used methods are the Kolmogorov-Smirnov test and the $\mathcal{X}^2$ test for numerical and categorical variables, respectively. For relational data, cardinality shape similarity is used, where for each parent row the number of child rows is calculated. This yields a numerical distribution for both real and synthetic data, on which a Kolmogorov-Smirnov test is performed.

#### 2.2.2 DISTANCE-BASED FIDELITY

Similar to statistical fidelity, distance-based fidelity is typically used to assess the fidelity of marginal distributions. However, some distance metrics also assess entire tables. Commonly used distance-based methods are total variation distance, Kullback-Leibler divergence, Jensen-Shannon distance, Wasserstein distance, maximum mean discrepancy, and pairwise correlation difference. Unlike statistical methods, reports of distance-based fidelity do not include hypothesis testing or any other quantification of uncertainty. This is an issue both when evaluating a method and when comparing two methods. In the former, a method can achieve a seemingly high distance that is in a high probability region when taking into account the sampling distribution. In the latter, a seemingly large difference between the two methods can be explained away by the variance of the sampling distribution.

#### 2.2.3 DETECTION-BASED FIDELITY

The basic idea of detection-based fidelity is to learn a model that can discriminate between real and synthetic data. If the model can achieve better-than-random predictive performance, this indicates that there are some patterns that identify synthetic data. Recent work by Zein & Urvoy (2022)

shows that using discriminative models can highlight the differences between real and synthetic tabular data.

The most common detection-based method is logistic detection (LD) (Gueye et al., 2023; Solatorio & Dupriez, 2023; Li & Tay, 2023; Pang et al., 2024), where a logistic regression model is used for discrimination. An extended version of LD known as parent-child logistic detection (P-C LD) is used to evaluate relational data. P-C LD applies LD to denormalized pairs of synthetic parent and child tables, assessing the preservation of parent-child relationships. A serious issue with denormalization is that it may introduce correlation between rows, breaking the i.i.d. assumption. This results in an over-performance of the discrimintative model and in underestimating the quality of the method for synthesizing relational data. It also makes it impossible to set a detection threshold for testing fidelity (for example, accuracy would be greater than 50% even if both datasets were from the same data generating process). For these reasons, we do not consider P-C detection.

Note that logistic regression is unable to capture interactions between columns unless these interactions are explicitly included as features. Therefore, LD is unable to discriminate between real and synthetic data when the marginal distributions are synthesized well. Furthermore, a mean-preserving transformation can produce synthetic data that LD will not be able to discriminate, although the synthetic data will be very different from the original data. We demonstrate this empirically in Appendix D.1. The popularity of LD implies a lenient evaluation of the state-of-the-art methods. Tree-based ensemble models are a better alternative, which is also suggested by the findings of Zein & Urvoy (2022) for tabular data.

### 2.2.4 MACHINE LEARNING UTILITY

The utility of synthetic data is most commonly measured with machine learning efficacy (ML-E) - comparing the hold-out performance of a predictive model trained on the original data with a predictive model trained on a synthetic dataset (Canale et al., 2022; Li & Tay, 2023; Mami et al., 2022; Solatorio & Dupriez, 2023; Pang et al., 2024). Patki et al. (2016) measured utility with a user study and Hansen et al. (2023) with the ability to retain model ranking or feature importance ranking (measured with rank correlation) in the train-on-synthetic evaluate-on-real paradigm. It is important to highlight that all these studies evaluated utility on a single-table, even those that investigated synthetic relational data.

Note that the typically used unweighted rank correlation (for example, Spearman or Kendall correlation coefficients) could be misleading. The issue gets worse as we increase the number of models or features, and their ordering becomes more susceptible to noise, especially among the models close to optimal performance and irrelevant features. That is, unweighted ranking will be most affected by the ranking of models and features in areas where ranking is of little practical utility.

### 2.3 RELATIONAL DATASETS AND SAMPLING PROCEDURES

We organize the datasets used in related work based on the structure of their relational schema, defined in Section 3. Datasets using only linear relationships (one parent and one child table) include AirBnB (Montoya et al., 2015) and Rossmann Store Sales (FlorianKnauer, 2015). While this structure may be sufficient for some practical applications, Gueye et al. (2023) and Xu et al. (2023) highlight the need for methods supporting more complex, multiple-parent relational structures found in datasets like *MovieLens* (Harper & Konstan, 2015) and *World Development Indicators* (World Bank, 2019). Datasets including multiple child tables include *Telstra Network Disruptions* (Wendy Kan, 2015), *Walmart Recruiting - Store Sales Forecasting* (Walmart, 2014), and *Mutagenesis* (Debnath et al., 1991). Datasets with multiple children and parents include *Coupon Purchase Prediction* (Kato et al., 2015), *World Development Indicators* (World Bank, 2019), *MovieLens* (Harper & Konstan, 2015) and *Biodegradability* (Blockeel et al., 1999). An additional possibility in relational databases is the use of composite foreign keys, which only the IRG (Xu et al., 2023) method supports. One such dataset is the *Grants* database (Alawini et al., 2018).

An important issue with evaluating relational data is that representative sampling is difficult (Buda et al., 2013; Gemulla et al., 2008). If the dataset does not include a time component or if the relationships are non-linear, the sampling becomes non-trivial and directly impacts the performance of the generative method. Even if the data have a strict hierarchy between tables, the rows in a child

table are related via their parent, which breaks the assumption of i.i.d. sampling. Typically, the method for synthesizing relational data is trained using the entire original dataset.

Note that a benchmark for relational learning based on graphs was recently proposed by Fey et al. (2023). It includes a collection of relational datasets along with machine learning tasks with defined train, evaluation, and test splits. However, these datasets include modalities such as text, which are not supported by the generative models evaluated in this work.

# 3 A General Approach to Fidelity with Discriminative Detection

In this section, we propose discriminative detection (DD), a generalization of the detection-based approach to fidelity, and its extension to relational data using aggregation (DDA). We are primarily motivated by the issues of existing approaches to multi-table fidelity, cardinality shape similarity (see Section 2.2.1) and P-C LD (see Section 2.2.3), and the subsequent need to strengthen the testing of this aspect in our benchmark. However, as we show, DD also improves on existing approaches to single-column and single-table fidelity.

Fidelity methods are concerned with measuring the similarity between two databases with the same schema but different data. Typically, these will be the real database $\mathbb{D}_{\text{REAL}}$ and a synthetic database $\mathbb{D}_{\text{SYN}}$, with the goal of detecting whether, to what extent, and where the synthetic data differ from the real data.

Let a relational database be a collection of tables $\mathcal{T} = \{T_1, ..., T_n\}$ and a schema $\mathcal{S} = (\mathcal{R}, \mathcal{A})$, where $\mathcal{R} \subseteq \mathcal{T} \times \mathcal{T}$ are the relations between the tables and $A_{T_i} = \{a_1^{T_i}, ..., a_l^{T_i}\} \in \mathcal{A}$ define the tables' attributes. Each table is a set $T = \{v_1, ..., v_{n_T}\}$ consisting of elements $v_i$ called rows. Each row $v \in T$ has three components $v = (p_v, \mathcal{K}_v, x_v)$. A **primary key** $p_v$ that uniquely identifies the row $v$; the set of **foreign keys** $\mathcal{K}_v = \{p_{v'} : v' \in T' \text{ and } (T, T') \in \mathcal{R}\}$, where $p_{v'}$ is the primary key of the row $v'$; and the set of **values** $x_v = \{(a, x) : a \in A_T\}$ corresponding to attributes of table $T$.

DD is summarized in Algorithm 1. It can be used for single-table, multi column, or single-column fidelity, which we determine by selecting target table and the subset of target columns. We then combine the two datasets and label the real and synthetic observations. From this point onwards, DD can be interpreted as a classification task of discriminating between real and synthetic observations. First, we use the selected classifier and error estimation procedure to estimate generalization accuracy. Then we use a Binomial test for proportion to test the deviation from baseline accuracy. Any better than random predictive performance implies a deviation from perfect fidelity. If a deviation is detected, we can optionally use an interpretability method to provide additional insight into where the classifier is able to distinguish between real and synthetic data.

In practice, we have to choose a classifier, an interpretability method for our classifier (optional), and a procedure for estimating accuracy. In our experiments we achieved good results with common choices of XGBoost, built-in feature importance, and cross-validation (see Section 4 for details). However, we could also consider multiple classifiers and perform model selection.

---

**Algorithm 1 Discriminative Detection**

---

**Require:** relational databases $\mathbb{D}_{\text{REAL}}$ and $\mathbb{D}_{\text{SYN}}$ that follow schema $\mathcal{S}$
**Require:** target table $T_i$ and target columns $\mathcal{I}$
**Require:** classifier $C$
**Require:** classification accuracy estimation method $M$
**Require:** (*optional) explainability method $E$

1: $T_{\text{REAL}} \leftarrow \text{SelectColumns}(\mathbb{D}_{\text{REAL}}, T_i, \mathcal{I})$      ▷ n rows
2: $T_{\text{SYN}} \leftarrow \text{SelectColumns}(\mathbb{D}_{\text{SYN}}, T_i, \mathcal{I})$      ▷ m rows
3: $X \leftarrow \begin{bmatrix} T_{\text{REAL}} \\ T_{\text{SYN}} \end{bmatrix}, y \leftarrow \begin{bmatrix} \vec{1} \in \mathbb{R}^{n \times 1} \\ \vec{0} \in \mathbb{R}^{m \times 1} \end{bmatrix}$
4: $loss_{0-1} \in \{0, 1\}^{n+m \times 1} \leftarrow \text{M}(C, X, y)$
5: $pval \leftarrow \text{BinomTest}(loss_{0-1}, \frac{max(n,m)}{n+m})$
6: return $loss_{0-1}, pval, ^*E(C, X, y)$

---

## 3.1 JUSTIFICATION AS A TWO SAMPLE TEST

DD can be interpreted as a null-hypothesis test for comparing two distributions (two sample testing) with classification accuracy as a proxy. The classifier serves as a map from high-dimensional data to a one-dimensional test statistic.

Using ML models is a common approach to two sample testing of high-dimensional data, with methods such as maximum mean discrepancy (Gretton et al., 2012) also used for single-table fidelity of synthetic data. Using predictive performance as a proxy is less common, but it has been receiving more attention (see Lopez-Paz & Oquab (2017), Kim et al. (2021) and Snoke et al. (2018)).

It has been shown that the accuracy-based approach to two-sample testing is consistent and controls for Type I error and (asymptotically) Type II error (see Kim et al. (2021) for theoretical results and a summary of empirical results). In practice, we are also interested in finite sample behavior. Experiment-based recommendations show that the approach should have an advantage in power when the data are well-structured or we have a lot of data, or when it is difficult to specify a test statistic, which is very common for high-dimensional data. Therefore, the large, higher-dimensional and structured nature of relational data is a perfect fit for DD.

## 3.2 MULTI-TABLE FIDELITY USING AGGREGATION

Discriminative detection with aggregation (DDA) extends DD to multi-table fidelity by augmenting the table with columns that aggregate information from child tables. Aggregation is an established technique in the field of relational reasoning (Getoor et al., 2007; Džeroski, 2010) and DDA can be thought of as a propositionalization (Kramer et al., 2001) approach to the C2ST on relational data. In DDA we replace the column selection in rows 1 and 2 of Algorithm 1 with calls to the aggregation algorithm described in Algorithm 2:

1: $T_{\text{REAL}} \leftarrow \text{RelationalAggregation}(\mathbb{D}_{\text{REAL}}, i)$
2: $T_{\text{SYN}} \leftarrow \text{RelationalAggregation}(\mathbb{D}_{\text{SYN}}, i)$

For each child table we add *CountRows*, a count of the the number of child rows corresponding to a parent row. For each attribute in each child table, we compute an aggregation attribute (*mean*, *count*, etc.). The aggregation attributes are added to the target table. In practice, different aggregation functions may be applied, as long as they maintain the i.i.d. assumption of the data.

## 3.3 ADVANTAGES OVER RELATED WORK

DD generalizes LD in two ways: by allowing for any classifier and by wrapping detection into a statistical testing framework that simplifies decision making. Aggregation (DDA) further generalizes the approach to multi-table fidelity and addresses the issues of the two methods that are commonly used for multi-table fidelity: cardinality shape similarity, which focuses on a very specific aspect, and Parent-Child LD, which suffers from the issues of denormalization.

DD (and DDA) can also be used to detect data copying. For example, if tables A and B are perfect copies, some training observations from A will have their corresponding copies in the test set of B (and vice versa). If any pattern is learned on these, the test data will have the labels reversed, and accuracy can drop below $\frac{1}{2}$. Instead of accuracy, LD typically uses $2 \cdot \max(\text{AUC}, \frac{1}{2}) - 1$, based on the popular implementation (DataCebo, 2022). Limiting to a minimal value of $\frac{1}{2}$ can therefore mask data copying (see Appendix D.2 for an example).

## 4 BENCHMARKING AND RESULTS

We combine our findings into a synthetic relational data benchmark, including single-column, single-table and multi-table fidelity metrics, as well as machine learning utility metrics.

We compare the following methods for synthesizing relational data: **SDV**, **RC-TGAN**, **REaLTab-Former**, and **ClavaDDPM**. Other related work does not have an API or available source code or we were not able to run the source code. We also included two of the most popular commercial tools, **MostlyAI** and **GretelAI**. The former do not disclose their generative method, while the latter provide two generative models **TabularLSTM** and **ACTGAN**. As a baseline for single-column and

single-table comparison, we also include state-of-the-art single-table methods (see Appendix D.5 for details).

We include 5 datasets that feature in related work (**AirBnB**, **Rossmann**, **Walmart**, **Biodegradability**, **MovieLens**) and the **Cora** dataset by McCallum et al. (2000), a popular dataset in graph representation learning. The datasets vary in types of relationships and number of tables and columns (see Appendix B.2 for details).

Our evaluation focuses on all three levels of synthetic relational data generation, with a focus on multi-table evaluation (see Appendix B.1 for details). For statistical metrics, confidence intervals and p-values are readily available, and for detection methods we use a binomial model. For distance-based metrics we use bootstrapping (1000 bootstrap replications) to approximate the sampling distribution. For the purposes of this evaluation, we consider a method failed to achieve fidelity if the difference between original and synthetic data is significant at level $\alpha = 0.05$. Most methods are non-deterministic, so we report results for three different replications. However, all results are stable across replications.

We use DD with either logistic regression or XGBoost and for DDA we augment the rows with (a) counts of child rows for each row in each parent table, (b) the mean values of the numeric columns in the child table corresponding to the parent row, and (c) the number of unique categories in related rows. We use 10-fold stratified cross-validation to estimate DD accuracy.

## 4.1 SINGLE-COLUMN PERFORMANCE

Single-column results show that most methods have trouble synthesizing even marginal distributions (see Table 9 in the Appendix). The diffusion-based approach ClavaDDPM performs better than the rest of the methods, but is limited by dataset structure (see Appendix A). Figure 1 shows how SDV, which models columns with simple predefined distributions performs poorly in most cases, while deep learning methods perform better. Despite this, SDV is still the primary baseline in related work, motivating our comprehensive comparison of current methods.

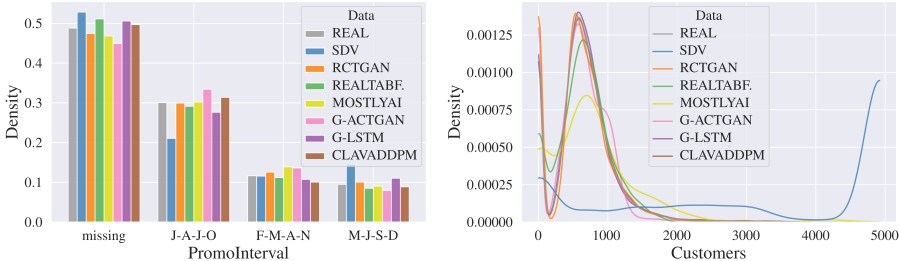

Figure 1: **Examples of marginal distributions on the Rossmann Dataset.** Deep learning-based methods generally synthesise both categorical and continuous marginal distributions well enough to pass the eye test. SDV, a commonly used baseline often fails to model even marginal distributions.

## 4.2 SINGLE-TABLE PERFORMANCE

The single-table results are worse than the single-column results (see Table 10 in the Appendix). In most cases methods fail the detection metric. Note that the relational synthetic data methods synthesize parent tables better than child tables (see Figure 2a). We hypothesize that this is due to the generation of child rows conditionally on parent rows, propagating errors down the hierarchy.

## 4.3 MULTI-TABLE PERFORMANCE

Multi-table metrics examine how well the referential integrity is preserved and how well the relationships between the columns of different tables are modeled. Cardinality shape similarity examines only the former and has fewer detections, while DD examines both. With few exceptions, methods fail to pass the multi-table fidelity tests based on detection (see Table 1). In the following two

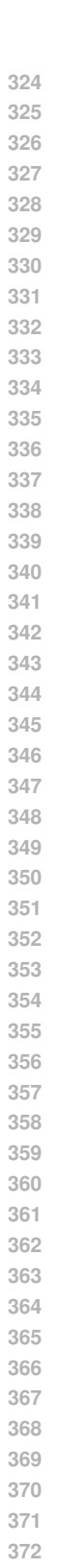
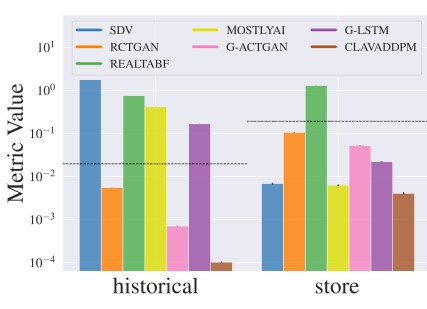
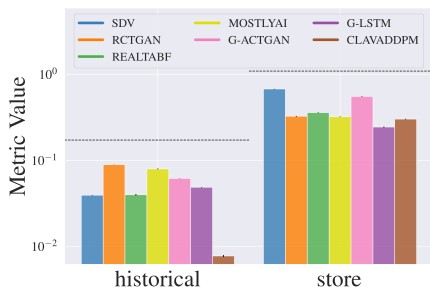

(a) Maximum Mean Discrepancy  (b) Pairwise Correlation Difference

Figure 2: **Maximum mean discrepancy (a) and pairwise correlation difference (b) on the Rossmann dataset.** The dotted line indicates the 95% bootstrapped confidence interval of the metric on original data. In Figure a, we observe that most methods model the parent table (store) better as the tests find more differences for the child table (historical). In Figure b however, although the metric values are higher for the parent table, the metric fails to detect differences in either table. This highlights the importance of interpreting a metric in the context of its uncertainty when analyzing the original data.

sections, we examine how DDA reveals shortcomings in relational fidelity, even in cases where single-table fidelity is preserved, and confirm this using interpretability methods.

Table 1: **Multi-table results.** We report the number of times the method failed the fidelity test. There are three numbers for each combination, one for each replication. The number in parentheses is the total number of tests per run. For cardinality shape similarity a test is run for every relationship in the dataset, while DD with aggregation is run for every table with dependent tables.

| Dataset | Method | Statistical | Detection | |
| --- | --- | --- | --- | --- |
| | | Cardinality | Agg LD | Agg XGB |
| AirBnB | SDV | 1, 1, 1 (1) | 1, 1, 1 (1) | 1, 1, 1 (1) |
| | RCTGAN | 1, 1, 1 (1) | 1, 1, 1 (1) | 1, 1, 1 (1) |
| | REALTABF. | 1, 1, 1 (1) | 1, 1, 1 (1) | 1, 1, 1 (1) |
| | MOSTLYAI | 1, 1, 1 (1) | 1, 1, 1 (1) | 1, 1, 1 (1) |
| | **G-ACTGAN** | **0, 0, 0 (1)** | 1, 1, 1 (1) | 1, 1, 1 (1) |
| | **G-LSTM** | **0, 0, 0 (1)** | 1, 1, 1 (1) | 1, 1, 1 (1) |
| | **CLAVADDPM** | **0, 0, 0 (1)** | 1, 1, 1 (1) | 1, 1, 1 (1) |
| Rossmann | SDV | **0, 0, 0 (1)** | 1, 1, 1 (1) | 1, 1, 1 (1) |
| | RCTGAN | 1, 1, 1 (1) | 1, 1, 1 (1) | 1, 1, 1 (1) |
| | REALTABF. | 1, 1, 1 (1) | 1, 1, 1 (1) | 1, 1, 1 (1) |
| | MOSTLYAI | 1, 1, 1 (1) | 1, 1, 1 (1) | 1, 1, 1 (1) |
| | G-ACTGAN | **0, 0, 0 (1)** | 1, 1, 1 (1) | 1, 1, 1 (1) |
| | G-LSTM | **0, 0, 0 (1)** | 1, 1, 1 (1) | 1, 1, 1 (1) |
| | **CLAVADDPM** | **0, 0, 0 (1)** | **1, 1, 0 (1)** | 1, 1, 1 (1) |
| Walmart | SDV | 1, 1, 1 (2) | 1, 1, 1 (1) | 1, 1, 1 (1) |
| | RCTGAN | 1, 0, 1 (2) | 1, 1, 1 (1) | 1, 1, 1 (1) |
| | REALTABF. | 2, 1, 1 (2) | 1, 1, 1 (1) | 1, 1, 1 (1) |
| | MOSTLYAI | 1, 1, 1 (2) | 1, 1, 1 (1) | 1, 1, 1 (1) |
| | **G-ACTGAN** | **0, 0, 0 (2)** | 1, 1, 1 (1) | 1, 1, 1 (1) |
| | **G-LSTM** | **0, 0, 0 (2)** | 1, 1, 1 (1) | 1, 1, 1 (1) |
| | **CLAVADDPM** | **0, 0, 0 (2)** | 1, 1, 1 (1) | 1, 1, 1 (1) |
| Biodeg. | SDV | 3, 3, 3 (4) | 3, 3, 3 (3) | 3, 3, 3 (3) |
| | RCTGAN | 4, 3, 3 (4) | 3, 2, 2 (3) | **3, 2, 3 (3)** |
| | MOSTLYAI | 4, 4, 4 (4) | 3, 3, 3 (3) | 3, 3, 3 (3) |
| | **G-ACTGAN** | **0, 0, 0 (4)** | **2, 2, 2 (3)** | 3, 3, 3 (3) |
| | **G-LSTM** | **0, 0, 0 (4)** | **2, 2, 2 (3)** | 3, 3, 3 (3) |
| MovieLens | RCTGAN | 6, 5, 5 (6) | 4, 3, 3 (4) | 4, 4, 4 (4) |
| | MOSTLYAI | 6, 6, 6 (6) | **3, 3, 3 (4)** | 4, 4, 4 (4) |
| | G-ACTGAN | **0, 0, 0 (6)** | 4, 4, 4 (4) | 4, 4, 4 (4) |
| | **CLAVADDPM** | **0, 0, 0 (6)** | 4, 4, 3 (4) | 4, 4, 4 (4) |
| CORA | SDV | 2, 2, 2 (2) | 1, 1, 1 (1) | 1, 1, 1 (1) |
| | RCTGAN | 1, 2, 2 (2) | 1, 1, 1 (1) | 1, 1, 1 (1) |
| | **G-ACTGAN** | **0, 0, 0 (2)** | 1, 1, 1 (1) | 1, 1, 1 (1) |
| | **G-LSTM** | **0, 0, 0 (2)** | 1, 1, 1 (1) | 1, 1, 1 (1) |

## 4.4 DISCRIMINATIVE DETECTION WITH AGGREGATION

Figure 3 shows that DD with XGBoost is able to better distinguish between real and synthetic data than LD on single-table fidelity. When incorporating the relational information by adding aggregations, the differences are more pronounced. Adding aggregations to LD allows it to detect a synthetic dataset even when marginal distributions for a single-table are perfectly generated (Fig. 3a) indicating that methods fail at preserving the characteristics of the relationships between tables. When using XGBoost as the discriminative model, the contribution of aggregations is similar, except in cases where discriminating between real and synthetic observations is already trivial (Fig. 3b). The best performing combination of DD with XGBoost and aggregation is, in almost all cases, able to identify a synthetically generated dataset and is often able to discriminate between individual observations with high accuracy. In particular, even when methods pass the single-table fidelity test (Fig. 3a, method G-LSTM), DDA reveals that the method fails to model relationships between columns in connected tables.

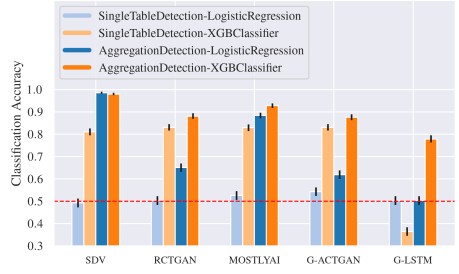

(a) Dataset Biodegradability, table Molecule

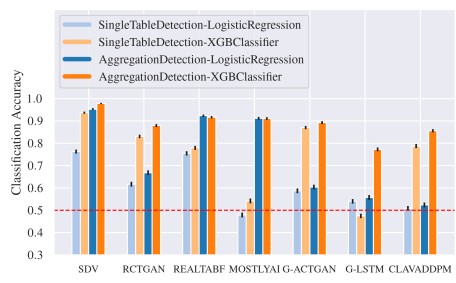

(b) Dataset Rossmann, table Store

Figure 3: **Discrimination accuracy for DD and DD with aggregation.** The results are for the parent tables. The red dashed line marks the expected 50% accuracy for perfectly generated data.

## 4.5 INTERPRETABILITY FOR GENERATIVE METHOD DIAGNOSTICS

ML interpretability with feature importance confirms that methods struggle with preserving the relationships between columns across tables. Figure 4 shows an example of how information about child columns is the most discriminative feature for two methods that pass single-table fidelity tests. We examine two such relationships in Figure 5. The partial dependence plots of the first and fourth most important features from Figure 4b show how subsets of both categorical (Fig. 5a) and numerical (Fig. 5b) features' conditional distributions are informative to the discriminative model.

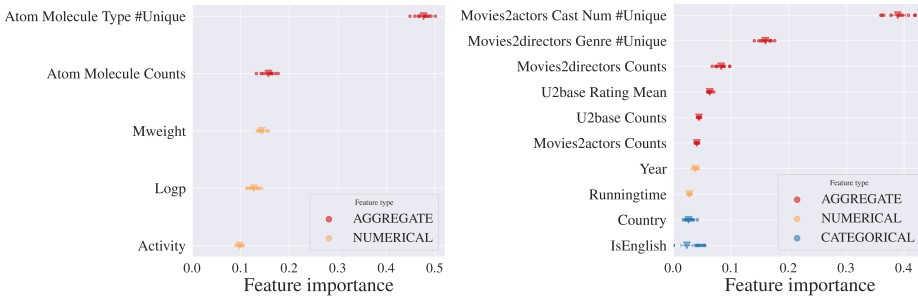

(a) Biodegradability - G-LSTM ($78\% \pm 1.6$)      (b) MovieLens - ClavaDDPM ($83.4\% \pm 0.4$)

Figure 4: **Feature importance for DD with aggregation using XGBoost.** Results are for the best performing methods (lowest DD accuracy). The added features that incorporate relational information (red) are the most important for discriminating between real and synthetic data. Notably, methods synthesize individual tables well, passing single-table fidelity tests in both cases.

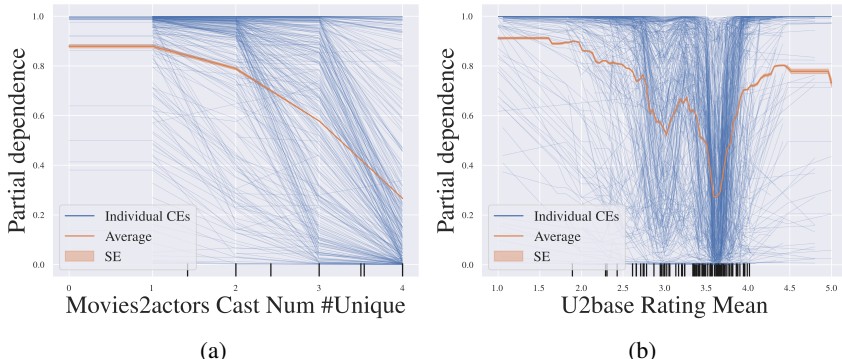

(a)                                     (b)

Figure 5: **Partial dependence plots**. Results are for the 1st and 4th most important feature from Figure 4b. With ideally generated synthetic data, features could not discriminate between synthetic and original data and every partial dependence plot would be a horizontal line at 50% probability. We can observe that (a) the synthetic data have too many unique actor cast numbers (higher probability of being synthetic when feature value is larger than 4) and (b) the mean movie ratings in the original data vary more than in the synthetic data, where they are more concentrated around 3.5.

## 4.6 RELATIONAL MACHINE LEARNING UTILITY PERFORMANCE

We construct ML utility pipelines for the three datasets: **AirBnB**, **Rossmann**, and **Walmart**. These datasets meet two criteria: all methods were able to generate data and they contain a temporal feature that allows us to split the data for evaluation on a held-out test set. We provide a detailed description of the utility pipelines in Appendix C.3.

Table 2 summarizes the utility results. For the AirBnB classification task, most methods have a moderate drop in predictive performance, except SDV and REALTABFORMER, that have near naive baseline performance. For the two regression tasks, the predictive performance when using synthetic data is at most near the naive baseline, often much worse. The only exception is SDV on the Walmart dataset, where the performance is better than when trained on original data.

Table 2: **Machine Learning Utility.** XGB Score is the predictive performance of an XGBoost model tested on original data (higher is better for AirBnB, lower is better for Rossmann and Walmart). As a baseline for comparison, we add original data performance and, in parentheses, performance if we predict the majority class or mean (naive baseline). The Model and Feature selection columns show the Spearman Rank Correlation between original and synthetic data model and feature ordering.

| Dataset | Method | XGB Score | Model Selection | Feature Selection |
|---|---|---|---|---|
| | Real Data | $0.72 \pm 0.001$ (0.5) | - | - |
| | SDV | $0.51 \pm 0.002$ | $-0.43 \pm 0.03$ | $0.01 \pm 0.01$ |
| | RCTGAN | $\mathbf{0.7 \pm 0.001}$ | $0.88 \pm 0.01$ | $0.01 \pm 0.01$ |
| AirBnB | REALTABF. | $0.54 \pm 0.001$ | $0.40 \pm 0.01$ | $-0.00 \pm 0.01$ |
| | MOSTLYAI | $\mathbf{0.7 \pm 0.001}$ | $\mathbf{0.98 \pm 0.01}$ | $\mathbf{0.09 \pm 0.01}$ |
| | GRE-ACTGAN | $\mathbf{0.7 \pm 0.001}$ | $0.69 \pm 0.01$ | $\mathbf{0.09 \pm 0.01}$ |
| | GRE-LSTM | $0.67 \pm 0.001$ | $0.64 \pm 0.01$ | $-0.04 \pm 0.01$ |
| | CLAVADDPM | $0.55 \pm 0.003$ | $0.42 \pm 0.02$ | $0.03 \pm 0.01$ |
| | Real Data | $81 \pm 0.9$ (345) | - | - |
| | SDV | $3406 \pm 20$ | $0.0 \pm 0.02$ | $-0.28 \pm 0.02$ |
| | RCTGAN | $321 \pm 0.6$ | $\mathbf{0.54 \pm 0.04}$ | $0.16 \pm 0.03$ |
| Rossmann | REALTABF. | $424 \pm 3$ | $-0.04 \pm 0.03$ | $-0.37 \pm 0.02$ |
| | MOSTLYAI | $464 \pm 5$ | $0.07 \pm 0.02$ | $0.23 \pm 0.02$ |
| | GRE-ACTGAN | $328 \pm 0.4$ | $-0.75 \pm 0.04$ | $0.13 \pm 0.03$ |
| | GRE-LSTM | $333 \pm 0.4$ | $-0.36 \pm 0.04$ | $\mathbf{0.31 \pm 0.02}$ |
| | CLAVADDPM | $\mathbf{269 \pm 1}$ | $0.46 \pm 0.03$ | $0.28 \pm 0.02$ |
| | Real Data | $6,117 \pm 103$ (7,697) | - | - |
| | SDV | $\mathbf{4,954 \pm 66}$ | $0.68 \pm 0.02$ | $0.14 \pm 0.03$ |
| | RCTGAN | $8,194 \pm 154$ | $0.11 \pm 0.04$ | $\mathbf{0.31 \pm 0.02}$ |
| Walmart | REALTABF. | $19,071 \pm 431$ | $-0.43 \pm 0.03$ | $0.20 \pm 0.02$ |
| | MOSTLYAI | $9,827 \pm 213$ | $0.18 \pm 0.04$ | $-0.24 \pm 0.03$ |
| | GRE-ACTGAN | $9,942 \pm 81$ | $-0.11 \pm 0.03$ | $-0.31 \pm 0.02$ |
| | GRE-LSTM | $12,382 \pm 81$ | $\mathbf{0.75 \pm 0.05}$ | $0.15 \pm 0.02$ |
| | CLAVADDPM | $8759 \pm 65$ | $0.30 \pm 0.04$ | $0.12 \pm 0.02$ |

Model selection ranking results do not show any simple pattern. Overall, the methods do not preserve the model rankings well, sometimes even reversing the ranking for a negative correlation. Results improve on average if we use weighted rank correlation (see Appendix D.4, Figure 7). Feature selection rankings results are similar and again improve if we use weighted rank correlation (see Appendix D.4, Figure 8). This suggests that most methods are better at ordering the top models (features) than all models (features). As these are usually of more interest than bottom performing models (least important features), unweighted rank correlation might not be the best approach.

Utility results suggest that the generated data might still be useful for certain tasks (e.g. developing classification pipelines on synthetic datasets) despite failing the fidelity tests. This is supported by the improvement in performance when using the weighted rankings and by most methods performing well on the AirBnB dataset. These results are in line with previous work (Hansen et al., 2023) indicating that fidelity and utility are inherently separate aspects of the quality of synthetic data.

## 5 CONCLUSION

We surveyed methods for synthesizing relational data and provided a critical review of approaches to evaluating the fidelity and utility of synthetic data. We integrated our findings into the first benchmark tailored to evaluating *relational* synthetic data (see Appendix B.3 for a comparison with related tools). Our work is available as a Python package (URL *anonymised and the work included as supplementary material*). that can be easily extended with new methods, metrics, and datasets.

We introduced DD, a generalization of detection-based approaches to fidelity, based on framing the problem as a classification task. Compared to commonly used statistical and distance-based approaches, we have the additional choices of classifier and, for multi-table fidelity, engineering additional features. However, empirical results show that DD outperforms other approaches even with basic additional features and XGBoost. The approach can be applied to single-column, multi column, single-table, or, with aggregation (DDA), multi-table fidelity. Worse-than-random performance of the discriminative model is also a viable diagnostic for data-copying. Finally, we demonstrate how, by explaining the predictions of the discriminative model, we can gain additional insights into which aspects of the original data were not synthesized well. We argue that DDA is a viable one-size-fits-all approach for investigating the fidelity of synthetic data.

We used our benchmark for the first comprehensive evaluation and comparison of the state-of-the-art methods for generating synthetic relational data. Methods are not yet able to generate synthetic relational data that is indistinguishable from original data. Most methods have problems with marginal distributions at least on some benchmark datasets and with single-tables on most datasets. None of the methods capture the relational properties of the original data, which results in relatively poor fidelity and utility. We highlight this as an important direction for future work on relational data synthesis (see Appendix 5.1 for limitations of the study and directions for future work).

### 5.1 LIMITATIONS AND FUTURE WORK

Our work focused on fidelity and utility, but not privacy. While we do briefly touch upon one aspect of privacy - data copying - we delegate the research of privacy metrics for synthetic relational data to future work. More work needs to be done in understanding the relationship between model quality and feature importance and practical utility. Unweighted rankings are flawed and it is not clear what weighting should be used or if metrics of this type are even a practically relevant utility measure. Finally, several aspects of synthetic data evaluation are limited by the difficulty of representative sampling. More work needs to be done in understanding the limitations and preparing new benchmark datasets or dataset generators.

Our results reveal significant gaps in multi-table fidelity. However, utility metrics on some datasets show performance comparable to real data, even when fidelity tests fail, highlighting the practical value of the generated data. To improve fidelity, future methods should focus on the relational aspects, with graph representation learning on relational data (Fey et al., 2023) showing promise for both generative modeling and a general approach for evaluating multi-table utility.

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

APPENDIX

## A   A SURVEY OF SYNTHETIC RELATIONAL DATA GENERATION METHODS

The **Synthetic Data Vault (SDV)** (Patki et al., 2016) introduced the first learning-based method for generating relational data. The method is based on the Hierarchical Modeling Algorithm (HMA) synthesizer, which is a multivariate version of the Gaussian Copula method. The method converts all columns to a predefined set of distributions and selects the best-fitting one. To learn dependencies, columns are converted to a standard normal before calculating the covariances. Tables are modeled with a recursive conditional parameter aggregation technique, which incorporates child table covariance and column distribution information into the parent table. The method requires the relational structure or metadata, which has since become a common practice.

The work of Mami et al. (2022) leverages the graph representation of relational data using **Graph Variational Autoencoders**. They focus on the case of one primary table connected by an identifier to an arbitrary number of secondary tables. The approach begins by transforming categorical, datetime, and numeric attributes into a normalised numeric format using an invertible function. Subsequently, all tables' attributes are merged into a single-table, where rows from each table are vertically concatenated. This merged table, along with an adjacency matrix based on foreign key relations, forms a homogeneous graph representation of the dataset. Message passing is then applied to this graph representation using gated recurrent units (GRU). Following the message passing phase, the data is processed through a variational autoencoder, which encodes the joined table and random samples are taken from its latent space. These samples are then decoded back to the data space.

**Composite Generative Models** (Canale et al., 2022) propose a generative framework based on codecs for modeling complex data structures, such as relational databases. They define a codec as a quadruplet: C = (E,D,S,L), consisting of an encoder E producing embeddings and intermediate contexts, a decoder D for distribution representation, a sampler S and loss function L. The authors define the following codecs: Categorical and Numerical Codecs for individual columns, while composite data types are encoded using Struct and List Codecs, allowing for relational data synthesis. They also propose a specific implementation using causal transformers as generative models.

The **Row Conditional-TGAN (RC-TGAN)** (Gueye et al., 2023) extends the conditional tabular GAN model (Xu et al., 2019) to relational data. RC-TGAN incorporates data from parent rows into the child table GAN model, allowing it to synthesise data conditionally on the connected parent table rows. The ability for conditional synthesis allows the method to handle various relationship schemas without additional processing. They enhance RC-TGAN to capture the influence of grandparent rows on their grandchild rows, preserving this connection even when the relationship information is not transferred by the parent table rows. Database synthesis is based on the row conditional generator of RC-TGAN model trained for each table. First, all parent tables are synthesised, followed by sampling the tables for which parents are already sampled. This allows using the synthesised parent rows as features when synthesizing child table rows.

The **Incremental Relational Generator (IRG)** (Li & Tay, 2023) uses GANs to incrementally fit and sample the relational dataset. They first define a topologically ordered sequence of tables in the dataset. Parent tables are modeled individually, while child tables undergo a three-step generation process. First, a potential context table is constructed by combining data from all related tables through join operations and aggregation. Then, the model predicts the number of child rows to be generated for each parent row, which they call its degree. They then extend the context table with corresponding degrees. Taking this table as context, they use a conditional synthetic tabular data generation model to generate the child table.

The **Realistic Relational and Tabular Transformer (REaLTabFormer)** (Solatorio & Dupriez, 2023) focuses on synthesizing single parent relational data and employs a GPT-2 encoder with a causal language model head to independently model the parent table. The encoder is frozen after training and used to conditionally model the child tables. Each child table requires a new conditional model, implemented as a sequence-to-sequence (Seq2Seq) transformer. The GPT-2 decoder with a causal language model head is trained to synthesise observations from the child table, accommodating arbitrary-length synthetic data conditioned on an input. While this method supports conditional synthesis of child rows, only one level is supported by this method.

Xu et al. (2023) propose a method for modeling many-to-many (M2M) datasets via random graph generation. They leverage a heterogeneous graph representation of the relational data and propose a factorization for modeling the graph representation incrementally. First, the edges of the graph are generated unconditionally using a random graph model. Second, one of the tables is generated conditionally on the topology of edges. One way to achieve such conditioning is by using a node embedding. Lastly, the remaining tables are generated using the conditional table model, which requires the generation of each node of the table based on the currently generated tables and all connections. They achieve this by using set embeddings to conditionally generate connected tables. The authors propose two variants using different conditional table models **BayesM2M** and **NeuralM2M**.

**Privacy-preserving graphical models with latent variables. (PrivLava)** (Cai et al., 2023) synthesizes relational databases with foreign key dependencies under differential privacy (DP). PrivLava models each foreign key in a relational schema as a separate graphical model, incorporating latent variables to capture inter-relational dependencies. Each entity in a child table associated with a parent table is modeled using a latent variable representing characteristics of the relationship. The approach handles foreign key relationships by treating them as a directed acyclic graph (DAG). It incrementally models the tables following a topological order, beginning with root tables and then moving on to tables that depend on them. This ensures that each synthetic row in child tables is conditionally generated based on latent features of related parent rows. Noise is injected at various stages to achieve DP guarantees.

The **Cluster Latent Variable guided Diffusion Probabilistic Models (ClavaDDPM)** (Pang et al., 2024) utilizes classifier-guided diffusion models, integrating clustering labels as intermediaries between tables connected by foreign-key relations. The authors first propose a model for generating a single parent-child relationship. The connection between the tables is modeled by a latent variable obtained using Gaussian Mixture Model clustering. ClavaDDPM learns a diffusion process on the joint parent and latent variable distribution, followed by training a latent variable classifier on the child table to guide the diffusion model for the child table. Additionally, it includes a model to estimate child group sizes, to preserve relation cardinality. The authors then extend this to more parent-child constraints through bottom-up modeling and address multi-parent scenarios by employing majority voting to mitigate potential clustering inconsistencies. Despite strong performance on our benchmark a key limitation of the method is its inability to generate datasets with multiple relationships between pairs of tables.

# B  SYNTHETIC RELATIONAL DATA GENERATION BENCHMARK

We provide our work as a Python package. The main goal of the package is the evaluation of the quality of synthetic relational data. We can compare multiple methods across multiple datasets with the *Benchmark* class or evaluate a single method on a single dataset with the *Report* class. All of the results of the benchmark are saved as JSON files and then parsed by our package for results summarization and visualization. The package is open source under the MIT license and can easily be extended with new methods, evaluation metrics, or datasets.

## B.1  EVALUATION METRICS

We list the evaluation metrics for data fidelity and utility currently supported in our benchmark in Table 3, based on the granularity of the data they evaluate.

We do not aggregate the values of the metrics over all tables and/or columns in the dataset, but rather report the results for all metrics. We believe that a single aggregated value does not give a good representation of the fidelity or utility of the synthetic data. We focus our evaluation of fidelity on the inseparability of the synthetic data from the original data (see Appendix B.1.2). We believe this gives a better insight into the quality of the data than just reporting metric values, which depend on the support of the values of the data we are evaluating.

Table 3: **Evaluation Metrics supported in the benchmark.**

|  | **Single-Column** | **Single-Table** | **Multi-Table** |
|---|---|---|---|
| **Statistical** | KS Test, $\mathcal{X}^2$ Test | / | cardinality shape similarity |
| **Distance** | Total Variation, Hellinger, Jensen-Shannon, Wasserstein | Maximum Mean Discrepancy, Pairwise Correlation Difference | / |
| **Detection** | Discriminative Detection | Discriminative Detection | Aggregation Detection, Parent-Child Detection |
| **Utility** | / | Single-Table ML-Utility | Relational ML-Utility |

### B.1.1 RELATIONAL AGGREGATION DETAILS

Algorithm 2 describes how aggregation attributes are constructed from values in related tables based on foreign key relationships. The algorithm defines a propositionalisation (Lachiche, 2010) of the relational dataset given a target table of interest $T_i$.

---

**Algorithm 2 Relational Aggregation**.

---

**Require:** relational database $\mathbb{D}$ with tables $\mathcal{T}$ and relational schema $\mathcal{S} = \{\mathcal{R}, \{A_{T_1} \ldots A_{T_n}\}\}$
**Require:** target table $T$
1:  aggregationAttributes $\leftarrow [\,]$
2:  **for** each $C_i \in \{C : (C, T) \in \mathcal{R}\}$ **do**
3:      $\mathbf{x}_{\text{count}}^{C_i} \leftarrow CountRows(C_i, T)$              $\triangleright$ count the rows in $C_i$ corresponding to rows in $T$
4:      aggregationAttributes.append($\mathbf{x}_{\text{count}}^{C_i}$)
5:      **for** each $a_j^{C_i} \in A_{C_i}$ **do**
6:          $\mathbf{x}_{a_j}^{C_i} \leftarrow Agg(C_i, a_j^{C_i}, T)$              $\triangleright$ calculate aggregation attribute
7:          aggregationAttributes.append($\mathbf{x}_{a_j}^{C_i}$)
8:      **end for**
9:  **end for**
10: $i \leftarrow 0$
11: **for** each $v \in T$ **do**
12:     $(p_v, \mathcal{K}_v, x_v) \leftarrow v$
13:     **for** each $\mathbf{a} \in$ aggregationAttributes **do**
14:         $x_v \leftarrow x_v \cup \{(\mathbf{a}.\text{name}, \mathbf{a}[i])\}$              $\triangleright$ add aggregation attribute and value to $x_v$
15:     **end for**
16:     $i \leftarrow i + 1$
17: **end for**
18: return $T_i$              $\triangleright$ final table with all aggregations

---

### B.1.2 SEPARABILITY OF SYNTHETIC AND ORIGINAL DATA

Statistical metrics report the underlying statistic and p-value. We decide if the metric was able to separate synthetic data from the original data if the p-value is less than the significance level $\alpha$, which in our case is 0.05.

Distance metrics must report the metric value, the support of the values the metric can obtain and the goal (minimization or maximization of the metric). Depending on the support and goal, a bootstrap confidence interval is constructed, which can be asymmetric depending on the support. The separability of the original and synthetic data is decided based on the $1 - \alpha$ confidence interval. If the metric value falls outside of the confidence interval, the metric is able to differ between real and synthetic data.

Detection metrics report the classification accuracy, however it can be replaced with any classification metric. The separability of the data is determined using a one-sided binomial test for proportions, assuming a probability parameter of $\frac{max(n,m)}{n+m}$ (where $n, m$ are numbers of rows for real and synthetic datasets respectively) for both groups, which indicates complete inseparability of the data.

## B.2 DATASETS

Table 4 summarizes the relational datasets used in our benchmark. Five datasets are from related work and we add the *Cora* dataset by McCallum et al. (2000), which contains a simple yet challenging relational schema. We include 2 datasets per hierarchy type to progressively add complexity in generation. The datasets used in our evaluation are diverse in terms of the number of columns, tables and relationships.

Table 4: **A summary of the 6 benchmark datasets.** The number of columns represents the number of non-id columns. The collection is diverse and covers all types of relational structures.

| Dataset Name | # Tables | # Rows | # Columns | # Relations | Hierarchy Type |
|---|---|---|---|---|---|
| Rossmann Store Sales | 2 | 59.085 | 16 | 1 | Linear |
| AirBnB | 2 | 57.217 | 20 | 1 | Linear |
| Walmart | 3 | 15.317 | 17 | 2 | Multi Child |
| Cora | 3 | 57.353 | 2 | 3 | Multi Child |
| Biodegradability | 5 | 21.895 | 6 | 5 | Multi Child & Parent |
| IMDB MovieLens | 7 | 1.249.411 | 14 | 6 | Multi Child & Parent |

The **AirBnB** (Airbnb, 2015) dataset includes user demographics, web session records, and summary statistics. It provides data about users' interactions with the platform, with the aim of predicting the most likely country of the users' next trip.

The **Biodegradability** dataset (Blockeel et al., 1999) comprises a collection of chemical structures, specifically 328 compounds, each labeled with its half-life for aerobic aqueous biodegradation. This dataset is intended for regression analysis, aiming to predict the biodegradation half-live activity based on the chemical features of the compounds.

The **Cora** dataset (McCallum et al., 2000) is a widely-used benchmark dataset in the field of graph representation learning. It consists of academic papers from various domains. The dataset consists of 2708 scientific publications classified into one of seven classes and their contents. The citation network consists of 5429 links.

The **IMDB MovieLens** dataset (Harper & Konstan, 2015) comprises information on movies, actors, directors, and users' film ratings. The dataset consists of seven tables, each containing at least one additional feature besides the primary and foreign keys.

The **Rossmann Store Sales** (FlorianKnauer, 2015) features historical sales data for 1115 Rossmann stores. The dataset consists of two tables connected by a single foreign key. This makes it the simplest type of relational dataset. The first table contains general information about the stores and the second contains sales-related data.

The **Walmart** dataset (Walmart, 2014) includes historical sales data for 45 Walmart stores across various regions. It includes numerical, date-time and categorical features across three connected tables *store*, *features* and *depts*. The dataset is from a Kaggle competition, with the task of predicting department-wide sales.

## B.3 COMPARISON WITH EXISTING EVALUATION TOOLS

The most popular and comprehensive package for evaluating tabular synthetic data is Synthcity (Qian et al., 2023a;b). It supports many statistical, privacy and detection-based (with several different models) metrics.

The only package that supports multi-table evaluation is SDMetrics (DataCebo, 2022). It includes multi-table metrics cardinality shape similarity and parent-child detection with logistic detection and support vector classifier. The package is not easy to extend and limits the adaptation of metrics. We re-implement detection metrics (discriminative detection, aggregation detection, and parent-child detection) to be used with an arbitrary classifier supporting the Scikit-learn classifier API (Pedregosa et al., 2011; Buitinck et al., 2013). In SDMetrics, the results of different metrics are aggregated into a single-value, which limits the comparison of individual metrics between the methods and datasets. We re-implement the distance and statistical metrics so that each statistic, p-value, and confidence interval is easy to access.

Our benchmark package can be easily extended with new methods, metrics, and datasets. The process for adding custom metrics and new datasets is described in *(URL anonymised and the work included as supplementary material)*.

### B.4 LICENSE AND PRIVACY

We obtain the datasets from the public SDV relational demo datasets repository (`https://docs.sdv.dev/sdv/single-table-data/data-preparation/loading-data`, accessed June 6th, 2024.). The SDV project is licensed under the Business Source License 1.1 (`https://github.com/sdv-dev/SDV?tab=License-1-ov-file#readme`, which allows use for research purposes. We manually check all of the data to ensure it does not include any personally identifiable information. Some of the datasets contain processed columns, including aggregations of numerical values and connected table rows (eg. nb_rows_in_{related table}). The authors of SDV (Patki et al., 2016) confirmed that these aggregations are not part of the original datasets, so we post-process all of the datasets to include only the columns found in their original form and update the metadata accordingly.

We adapt some of the metrics from the SDMetrics (DataCebo, 2022) (MIT License) and Synthcity (Qian et al., 2023a;b) (Apache-2.0 License) synthetic data generation benchmarks.

## C EXPERIMENTS

### C.1 COMPUTATIONAL RESOURCES

The generative methods were trained on NVIDIA 32GB V100S GPUs and H100 80GB GPUs. The total number of GPU hours spent across all experiments is approximately 500. Results which do not require a GPU were run on machines running AMD EPYC 7702P 64-Core Processor with 256GB of RAM. All experiments were performed on an internal HPC cluster.

### C.2 REPRODUCIBILITY

#### C.2.1 DATASETS AND DATA SPLITTING

Scripts for downloading the datasets and their metadata in the SDV format (Patki et al., 2016) are available in the project repository *(URL anonymised and the work included as supplementary material)*, as well as the corresponding synthetic data samples for all methods to enable the reproduction of the benchmark results.

We opt not to split the datasets into train, test, and validation sets for generative model training. When no temporal information is included and the structure is non-linear the representative sampling in relational datasets is non-trivial. We delegate this to future work.

Due to computational limits (also reported by Solatorio & Dupriez (2023)), we subsample the Rossmann Store Sales, AirBnB, and Walmart datasets. The linear structure of these datasets allows us to representatively sample from them and split them temporally for the purposes of ML-Utility experiments. We subsample the datasets and use the remaining data to obtain hold-out test sets:

- **Rossmann Store Sales**: Subsampled on table *historical*, column *Date* by taking the rows of a two month period from *2014-07-31* to *2014-09-30*, similarly to Solatorio & Dupriez (2023).
- **AirBnB**: Subsampled the dataset by only including the users who have less than 50 sessions and then sampled 10k users, as done by Solatorio & Dupriez (2023).
- **Walmart**: Subsampled on tables *departments* and *features* on the column *Date* by taking the rows from December 2011.

#### C.2.2 EXPERIMENTAL DETAILS AND HYPERPARAMETERS

To provide some quantification of the variability from the non-deterministic nature of the methods, we generated synthetic data for each of the methods for each of the datasets 3 times with different fixed random seeds. We ran the benchmark for each replication.

Scripts for reproducing the generative model training and instructions for training commercial methods are included in the project repository.

It is possible that better performance could be achieved by investing more effort into parameter tuning. However, due to our choice to not split the data, it was not clear how to optimize hyperparameters; therefore, we selected default hyperparameters for all methods (see Table 5).

Table 5: **Hyperparameter specification.**

| model | hyperparameter | value |
|---|---|---|
| RCTGAN | embedding_dim | 128 |
| | generator_dim | (256, 256) |
| | discriminator_dim | (256, 256) |
| | generator_lr | 0.0002 |
| | generator_decay | 1e-06 |
| | discriminator_lr | 0.0002 |
| | discriminator_decay | 1e-06 |
| | batch_size | 500 |
| | discriminator_steps | 1 |
| | epochs | 1000 |
| | pac | 10 |
| | grand_parent | True |
| | field_transformers | None |
| | constraints | None |
| | rounding | "auto" |
| | min_value | "auto" |
| | max_value | "auto" |
| SDV | locales | None |
| | verbose | True |
| | table_synthesizer | "GaussianCopulaSynthesizer" |
| | enforce_min_max_values | True |
| | enforce_rounding | True |
| | numerical_distributions | {} |
| | default_distribution | "beta" |
| REALTABFORMER | epochs | 100 |
| | batch_size | 8 |
| | train_size | 0.95 |
| | output_max_length | 512 |
| | early_stopping_patience | 5 |
| | early_stopping_threshold | 0 |
| | mask_rate | 0 |
| | numeric_nparts | 1 |
| | numeric_precision | 4 |
| | numeric_max_len | 10 |
| | evaluation_strategy | "steps" |
| | metric_for_best_model | "loss" |
| | gradient_accumulation_steps | 4 |
| | remove_unused_columns | True |
| | logging_steps | 100 |
| | save_steps | 100 |
| | eval_steps | 100 |
| | load_best_model_at_end | True |
| | save_total_limit | 6 |
| | optim | "adamw_torch" |
| MOSTLYAI | Configuration presets | Accuracy |
| | Max sample size | 100% |
| | Model size | Large |
| | Batch size | Auto |
| | Flexible generation | Off |
| | Value protection | Off |
| G-LSTM | model | "synthetics/tabular-lstm" |
| | type | "gretel_tabular" |
| | num_records_multiplier | 1.0 |
| G-ACTGAN | model | "synthetics/tabular-actgan" |
| | type | "gretel_tabular" |
| | num_records_multiplier | 1.0 |
| CLAVADDPM | num_clusters | 50 |
| | parent_scale | 1.0 |
| | classifier_scale | 1.0 |
| | num_timesteps | 2000 |
| | batch_size | 4096 |
| | layers_diffusion | [512, 1024, 1024, 1024, 1024, 512] |
| | iterations_diffusion | 200000 |
| | lr_diffusion | 0.0006 |
| | weight_decay_diffusion | 1e-05 |
| | scheduler_diffusion | "cosine" |
| | layers_classifier | [128, 256, 512, 1024, 512, 256, 128] |
| | iterations_classifier | 20000 |
| | lr_classifier | 0.0001 |
| | dim_t | 128 |

### C.3 MACHINE LEARNING UTILITY PIPELINES

To include the relational aspect of the data, we incorporate the data from all tables using appropriate aggregations and table joins. For each dataset, we select the target column, which is most commonly used for prediction, and transform it where appropriate. The code for the pipelines is available in the benchmark repository: *(URL anonymised and the work included as supplementary material).*

For AirBnB we select Country Destination, the country of the user's first booking. As this is a highly imbalanced column, we simplify the task to determine whether a user will book a trip or not ($country\_destination \neq NDF$).

For the Rossmann dataset, the original target column is Sales. However, as the version of the dataset we use does not contain it, we select the Customers column, describing the number of customers visiting a store on a single day. Due to the size of the dataset, we aggregate the customer data to predict the monthly average number of customers for each store.

In the Walmart dataset, the target column, Weekly Sales, represents the sales for an individual department each week. The predictive task involves forecasting these weekly department-wide sales for each store.

For each dataset and generative method we fit multiple learners (XGBoost, Linear Regression, Random Forest, Decision Tree, K-Nearest Neighbors, Support Vector Machine, Gaussian Naive Bayes and Multi-Layer Perceptron). Each learner is trained twice, once on the original data and once on the synthetic data and evaluated on the held-out test set. We then compare the performance of the learners trained on the real and synthetic data.

Additionally, we also evaluate the generative models' ability to preserve the ranking of the learners and the rankings of features between the real and synthetic data. The test data is obtained from the rows unused during subsampling of the datasets. For the Rossmann and Walmart datasets we select the data for the next month after subsampling (November 2014 and January 2012 respectively). For the AirBnB dataset we randomly sample 2000 users that meet the same criteria as the training set (having at most 50 sessions).

On the Rossmann dataset we first join the Store and Historical tables based on the foreign keys, we then drop the State Holiday column which is constant in the training set and the Day of Week column as we aggregate the data by month. We then one hot encode the categorical columns and aggregate the data by Store, Month and Year. In this way we obtain the expected value for each of the store's numerical attributes and the expected frequencies for each column. Lastly, we impute the missing values with zeroes.

On the AirBnB dataset we first drop the Date of First Booking column as it can be used to perfectly predict the target. We then fill the missing numerical values with zeroes. We aggregate the average session duration and count the number of sessions for each user. We then add these values to the columns in the user table and use zeroes for the users with zero logged sessions. As described previously, we convert the country destination to a binary attribute, indicating whether a user made a reservation or not. This is mainly done due to the target column having a highly imbalanced distribution, which was an issue for all of the generative methods.

On the Walmart Dataset we simply join the Department and Store tables based on the Store id. We then merge it with the Features table on the Store id and Date columns. We then aggregate the data by Store and Date to obtain average Weekly sales for a store across all departments.

## D ADDITIONAL EXPERIMENTS

### D.1 SHORTCOMINGS OF LOGISTIC DETECTION

As explained in Section 2.2.3 a significant limitation of LD is its inability to capture interactions between columns. It can thus assign a perfect fidelity score to a dataset that is completely corrupted. In this section, we empirically show this shortcoming. We conduct the experiment by selecting a table from each dataset (with the exception of CORA in which no table has two columns, which are not primary or relational keys). We first select the parent table Stores and split in half to simulate the original table and a perfectly generated (by the underlying DGP) synthetic table. We then copy

the "generated" table and randomly shuffle values in each column, completely ruining the fidelity of the dataset, while keeping the marginal distributions intact. We then evaluate the perfectly generated and shuffled datasets using LD and DD using XGBoost. The results are visualized in Figure 6.

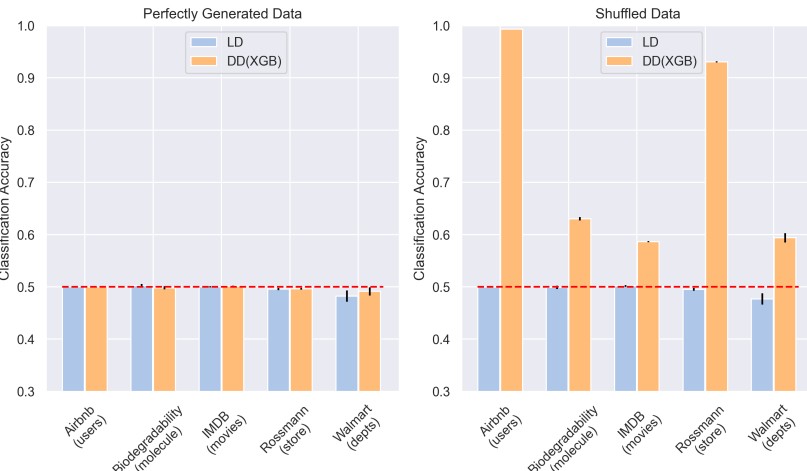

Figure 6: **Issues with logistic detection.** For each dataset, we simulate a perfectly generated table by splitting the original table in half. We copy one part of the table and shuffle the values in each column and thus completely ruin the fidelity of the table. While the DD metric using an XGBoost classifier can almost perfectly segment the corrupted rows, logistic regression assigns both of the datasets the same score.

Notably LD assigns both versions of the dataset the same score, labeling them indistinctive from the original data. If the fidelity aspect of interest would be solely the marginal distributions, the LD results would be more appropriate than those of DD using XGBoost (as marginals are identical in both datasets). However, given that we are interested in single-table fidelity, our experiment showcases a fundamental shortcoming of LD as a measure of single-table fidelity.

### D.2    DISCRIMINATIVE DETECTION AS A DATA COPYING DIAGNOSTIC

In this section we investigate how discriminative detection can be used to diagnose data copying. We also demonstrate how the classifier performance commonly reported in LD ($2 \cdot \max(\text{AUC}, \frac{1}{2}) - 1$) masks this issue. As in the previous experiment we simulate a perfect synthetic generating a dataset by splitting the original table in half. However, instead of introducing corruption into the second half, we create an exact copy of the original data (i.e., the first half). The commonly used LD implementation fails to detect data copying and assigns the copied data a perfect score. In contrast, DD successfully detects data copying as accuracy drops significantly below 50%.

We then examine the behaviour of DD when only a portion of the data is copied. We keep a portion of the dataset as an identical copy and sample the rest of the values from the "perfectly generated" half. For most of the datasets, even when a relatively low percentage of the data is copied, DD detects the duplication. We showcase the results in Figure 7.

### D.3    FIDELITY - UTILITY CORRELATION

We examine the relationship between fidelity and utility metrics. We compute the average utility score for each ML model used in the utility task. We then compare those with the fidelity score for discriminative detection with aggregation using an XGBoost model and logistic detection. We pair the average utility score with the detection accuracy for each generative method for each of the three replications. We then use bootstrap to estimate the correlation using $10,000$ replications for both fidelity metrics. We report the results in Table 6.

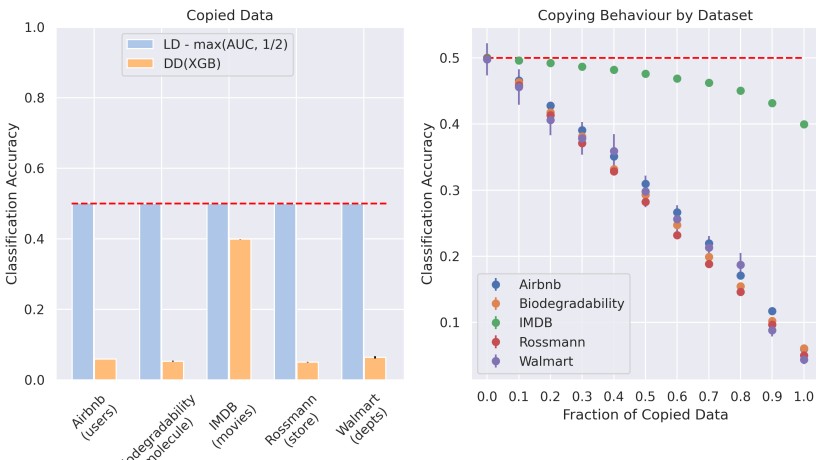

Figure 7: **Detecting data copying with DD.** The left plot demonstrates how the error estimation of LD ($2 \cdot \max(\text{AUC}, \frac{1}{2}) - 1$) masks data copying, while DD detects it across all datasets. In the right plot, we observe how copying only a fraction of the original data affects DD accuracy, with accuracy consistently decreasing as more data is duplicated.

| Dataset | $\rho(DDA_{XGB}, U)$ | $\rho(LD, U)$ | $\rho_{DDA} - \rho_{LD}$ |
|---------|---------------------|---------------|--------------------------|
| Rossmann | $-0.61$ | $-0.33$ | $-0.28(-0.43, -0.13)$ |
| Walmart | $-0.417$ | $0.03$ | $-0.45(-0.78, -0.07)$ |
| Airbnb | $-0.45$ | $-0.52$ | $0.07(0.02, 0.07)$ |
| Total | $-0.535$ | $-0.457$ | $-0.08(-0.22, 0.06)$ |

Table 6: **Detection - utility score correlation** comparison for DD with aggregation when using XGBoost and logistic detection. The estimated correlation for both models is negative, indicating an inverse relationship between a higher detection score (lower fidelity) and higher utility score. On average the utility score for DDA is lower than for LD indicating a stronger relationship.

On two of the tested datasets the utility score for DDA is lower than for LD implying a stronger relationship, with the exception being the Airbnb dataset. On this dataset, most methods struggle with generating the marginal distributions, resulting in both metrics achieving a high detection accuracy ($99.7 \pm 0.1\%$ and $96.1 \pm 2$ %respectively). LD achieves a significantly lower accuracy on RCTGAN ($74 \pm 0.4$ % as opposed to $98 \pm 0.03$ %). As RCTGAN scores best in utility, this causes a slightly higher correlation for LD.

### D.4 Weighted Model & Feature Ranking

As mentioned in Section 4.6 in a practical scenario one is more interested in a subset of the evaluated models and feature importances. When evaluating the utility of a generative method it makes sense to penalize the switches between unimportant features less. For this reason we also compute the weighted Kendall's $\tau$ alongside the Spearman and Kendall's $\tau$ rank correlation. Tables 7 and 8 show the difference in model and feature selection scores when using the weighted metric.

### D.5 Comparison with Single-Table Methods

In our single-column and single-table benchmarks, we include five state-of-the-art tabular generative methods included in the *Synthcity* library: Bayesian Networks (**BN**) (Ankan & Panda, 2015), Conditional Tabular GAN (**CTGAN**) (Xu et al., 2019), Tabular Diffusion Denoising Probabilistic Model (**TabDDPM**) (Kotelnikov et al., 2022), RQ-Neural Spine Flows (**NFLOW**) (Durkan et al., 2019), and Tabular Variational Autoencoder (**TVAE**) (Xu et al., 2019). We use hyperparameters that were used in the single-table evaluation of these methods by Hansen et al. (2023).

Table 7: **Model Rank:** Spearman vs. $\tau$ vs. Weighted $\tau$

| Dataset | Method | Spearman | Kendall | Weighted |
|---|---|---|---|---|
| AirBnB | SDV | $-0.43 \pm 0.03$ | $-0.29 \pm 0.03$ | $-0.08 \pm 0.02$ |
| | RCTGAN | $0.88 \pm 0.01$ | $0.71 \pm 0.01$ | $0.80 \pm 0.01$ |
| | REALTABF. | $0.40 \pm 0.01$ | $0.36 \pm 0.01$ | $0.49 \pm 0.02$ |
| | MOSTLYAI | $\mathbf{0.98 \pm 0.01}$ | $\mathbf{0.93 \pm 0.02}$ | $\mathbf{0.95 \pm 0.01}$ |
| | GRE-ACTGAN | $0.69 \pm 0.01$ | $0.57 \pm 0.01$ | $0.68 \pm 0.01$ |
| | GRE-LSTM | $0.64 \pm 0.01$ | $0.43 \pm 0.01$ | $0.71 \pm 0.01$ |
| | CLAVADDPM | $0.42 \pm 0.02$ | $0.29 \pm 0.02$ | $0.18 \pm 0.005$ |
| Rossmann | SDV | $0.0 \pm 0.02$ | $0.05 \pm 0.02$ | $-0.37 \pm 0.01$ |
| | RCTGAN | $\mathbf{0.54 \pm 0.04}$ | $\mathbf{0.43 \pm 0.03}$ | $\mathbf{0.78 \pm 0.03}$ |
| | REALTABF. | $-0.04 \pm 0.03$ | $0.05 \pm 0.02$ | $0.53 \pm 0.02$ |
| | MOSTLYAI | $0.07 \pm 0.02$ | $0.05 \pm 0.02$ | $-0.44 \pm 0.02$ |
| | G-ACTGAN | $-0.75 \pm 0.04$ | $-0.62 \pm 0.03$ | $0.35 \pm 0.01$ |
| | G-LSTM | $-0.36 \pm 0.04$ | $-0.24 \pm 0.03$ | $-0.26 \pm 0.03$ |
| | CLAVADDPM | $0.46 \pm 0.03$ | $0.41 \pm 0.02$ | $0.70 \pm 0.01$ |
| Walmart | SDV | $0.68 \pm 0.02$ | $0.52 \pm 0.02$ | $\mathbf{0.93 \pm 0.02}$ |
| | RCTGAN | $0.11 \pm 0.04$ | $0.14 \pm 0.03$ | $0.58 \pm 0.03$ |
| | REALTABF. | $-0.43 \pm 0.03$ | $-0.33 \pm 0.02$ | $0.1 \pm 0.01$ |
| | MOSTLYAI | $0.18 \pm 0.04$ | $0.05 \pm 0.03$ | $0.1 \pm 0.01$ |
| | G-ACTGAN | $-0.11 \pm 0.03$ | $-0.14 \pm 0.02$ | $0.36 \pm 0.02$ |
| | G-LSTM | $\mathbf{0.75 \pm 0.05}$ | $\mathbf{0.62 \pm 0.04}$ | $0.48 \pm 0.02$ |
| | CLAVADDPM | $0.30 \pm 0.04$ | $0.22 \pm 0.03$ | $0.41 \pm 0.01$ |

Table 8: **Features Rank:** Spearman vs. $\tau$ vs. Weighted $\tau$

| Dataset | Method | Spearman | Kendall | Weighted |
|---|---|---|---|---|
| AirBnB | SDV | $0.01 \pm 0.01$ | $0.01 \pm 0.01$ | $0.11 \pm 0.01$ |
| | RCTGAN | $0.01 \pm 0.01$ | $0.01 \pm 0.01$ | $0.62 \pm 0.00$ |
| | REALTABF. | $0.0 \pm 0.01$ | $0.0 \pm 0.01$ | $0.42 \pm 0.01$ |
| | MOSTLYAI | $\mathbf{0.09 \pm 0.01}$ | $\mathbf{0.07 \pm 0.01}$ | $\mathbf{0.7 \pm 0.003}$ |
| | GRE-ACTGAN | $\mathbf{0.09 \pm 0.01}$ | $0.06 \pm 0.01$ | $0.66 \pm 0.00$ |
| | GRE-LSTM | $-0.04 \pm 0.01$ | $-0.02 \pm 0.01$ | $0.53 \pm 0.01$ |
| | CLAVADDPM | $0.03 \pm 0.01$ | $0.02 \pm 0.01$ | $\mathbf{0.7 \pm 0.003}$ |
| Rossmann | SDV | $-0.28 \pm 0.02$ | $-0.18 \pm 0.02$ | $-0.11 \pm 0.02$ |
| | RCTGAN | $0.16 \pm 0.03$ | $0.16 \pm 0.02$ | $0.38 \pm 0.01$ |
| | REALTABF. | $-0.37 \pm 0.02$ | $-0.25 \pm 0.01$ | $0.31 \pm 0.02$ |
| | MOSTLYAI | $0.23 \pm 0.02$ | $0.16 \pm 0.01$ | $0.09 \pm 0.02$ |
| | G-ACTGAN | $0.13 \pm 0.03$ | $0.15 \pm 0.02$ | $0.3 \pm 0.03$ |
| | G-LSTM | $\mathbf{0.31 \pm 0.02}$ | $\mathbf{0.26 \pm 0.02}$ | $-0.28 \pm 0.02$ |
| | CLAVADDPM | $0.28 \pm 0.02$ | $0.20 \pm 0.02$ | $\mathbf{0.67 \pm 0.01}$ |
| Walmart | SDV | $0.14 \pm 0.03$ | $0.08 \pm 0.02$ | $-0.17 \pm 0.03$ |
| | RCTGAN | $\mathbf{0.31 \pm 0.02}$ | $\mathbf{0.21 \pm 0.01}$ | $0.27 \pm 0.03$ |
| | REALTABF. | $0.2 \pm 0.02$ | $0.12 \pm 0.02$ | $-0.1 \pm 0.02$ |
| | MOSTLYAI | $-0.24 \pm 0.03$ | $-0.16 \pm 0.02$ | $0.29 \pm 0.03$ |
| | G-ACTGAN | $-0.31 \pm 0.02$ | $-0.3 \pm 0.02$ | $0.17 \pm 0.03$ |
| | G-LSTM | $0.15 \pm 0.02$ | $0.08 \pm 0.02$ | $\mathbf{0.36 \pm 0.02}$ |
| | CLAVADDPM | $0.12 \pm 0.02$ | $0.09 \pm 0.02$ | $-0.09 \pm 0.01$ |

The results for single-column synthesis are shown in Table 9. We observe that the methods for relational data synthesis perform comparably to the tabular generative methods.

As expected, the performance of the methods degrades when modeling individual tables, which can be seen in Table 10. Here we observe a similar drop in performance for relational and single-table methods, with the methods that generated marginal distributions well also achieving better results in modeling whole tables.

We note that some of the methods either timed out (generation time was longer than 48 hours) or were not able to generate all of the tables of a particular dataset so they are not included in the Table 9 or Table 10.

Table 9: **Single-Column Results.** We report the number of times the method failed the fidelity test. There are three numbers for each combination, one for each replication. The number in parentheses is the total number of eligible columns for the corresponding metric.

| Dataset | Method | Statistical | | Distance | | | | Detection | |
|---|---|---|---|---|---|---|---|---|---|
| | | $\chi^2$ | KS | Hel. | JS | TV | Was. | LD | XGB |
| AirBnB | SDV | 15, 15, 15 (15) | 5, 5, 5 (5) | 13, 13, 13 (20) | 13, 13, 13 (20) | 14, 14, 14 (20) | **0, 0, 0 (5)** | 19, 19, 19 (20) | 20, 20, 20 (20) |
| | RCTGAN | 15, 15, 15 (15) | 5, 5, 5 (5) | 6, 6, 6 (20) | 7, 6, 6 (20) | 9, 10, 9 (20) | **0, 0, 0 (5)** | 18, 19, 18 (20) | 20, 19, 20 (20) |
| | REALTABF. | 15, 15, 15 (15) | 5, 4, 4 (5) | 15, 14, 15 (20) | 15, 14, 15 (20) | 15, 15, 14 (20) | 3, 2, 3 (4) | 19, 15, 16 (20) | 20, 17, 16 (20) |
| | MOSTLYAI | 12, 9, 8 (15) | **3, 1, 0 (5)** | 4, 5, 5 (20) | 4, 5, 5 (20) | 4, 5, 5 (20) | **0, 0, 0 (5)** | 15, 12, 11 (20) | 15, 11, 10 (20) |
| | G-ACTGAN | 14, 14, 15 (15) | 5, 5, 5 (5) | 2, 2, 1 (20) | 2, 2, 1 (20) | 6, 6, 7 (20) | 0, 0, 1 (5) | 18, 18, 18 (20) | 19, 19, 20 (20) |
| | G-LSTM | 15, 15, 15 (15) | 5, 3, 5 (5) | **1, 1, 1 (20)** | **1, 1, 1 (20)** | 2, 2, 3 (20) | **0, 0, 0 (5)** | 17, 15, 17 (20) | 18, 18, 19 (20) |
| | **CLAVADDPM** | **8, 7, 8 (15)** | 3, 3, 3 (5) | **1, 1, 1 (20)** | **1, 1, 1 (20)** | **1, 1, 1 (20)** | 1, 1, 1 (5) | **6, 6, 5 (20)** | **8, 8, 7 (20)** |
| | BN | 9, 9, 9 (15) | 3, 3, 3 (5) | 7, 7, 7 (20) | 7, 7, 7 (20) | 7, 7, 7 (20) | **0, 0, 0 (5)** | 12, 12, 12 (20) | 14, 14, 14 (20) |
| | CTGAN | 15, 15, 15 (15) | 5, 5, 5 (5) | 8, 8, 8 (20) | 8, 8, 8 (20) | 9, 10, 8 (20) | **0, 0, 0 (5)** | 19, 18, 18 (20) | 19, 20, 20 (20) |
| | DDPM | 15, 15, 13 (15) | 5, 5, 4 (5) | 2, 5, 2 (20) | 2, 2, 2 (20) | 3, 5, 3 (20) | **0, 0, 0 (5)** | 17, 18, 16 (20) | 19, 20, 18 (20) |
| | NFLOW | 15, 15, 15 (15) | 5, 5, 5 (5) | 18, 18, 13 (20) | 18, 18, 13 (20) | 18, 18, 14 (20) | 3, 3, 1 (5) | 19, 19, 18 (20) | 20, 20, 20 (20) |
| | TVAE | 14, 14, 14 (15) | 5, 5, 5 (5) | 8, 8, 8 (20) | 8, 8, 8 (20) | 8, 8, 8 (20) | **0, 0, 0 (5)** | 18, 18, 18 (20) | 19, 19, 19 (20) |
| Rossmann | SDV | 7, 7, 7 (9) | 7, 7, 7 (7) | 7, 7, 7 (16) | 7, 7, 7 (16) | 7, 7, 7 (16) | 1, 1, 1 (7) | 9, 9, 9 (16) | 14, 14, 14 (16) |
| | RCTGAN | 6, 6, 6 (9) | 6, 7, 6 (7) | **1, 0, 0 (16)** | **1, 0, 0 (16)** | 2, 2, 3 (16) | **0, 0, 0 (7)** | 10, 11, 11 (16) | 12, 13, 12 (16) |
| | REALTABF. | 4, 3, 3 (9) | **2, 2, 2 (7)** | 2, 3, 2 (16) | 2, 3, 2 (16) | 3, 3, 2 (16) | 1, 1, 1 (7) | 7, 6, 4 (16) | **8, 6, 5 (16)** |
| | MOSTLYAI | 4, 5, 5 (9) | **2, 2, 2 (7)** | 2, 4, 2 (16) | 2, 4, 2 (16) | 3, 5, 3 (16) | 1, 1, 1 (7) | 6, 7, 7 (16) | 6, 9, 8 (16) |
| | G-ACTGAN | 4, 6, 6 (9) | 7, 7, 6 (7) | 1, 0, 1 (16) | 1, 0, 1 (16) | 2, 3, 2 (16) | **0, 0, 0 (7)** | 9, 11, 7 (16) | 11, 13, 13 (16) |
| | G-LSTM | 6, 6, 6 (9) | 3, 3, 3 (7) | 0, 0, 3 (16) | 0, 0, 3 (16) | **0, 0, 5 (16)** | 0, 0, 1 (7) | 9, 8, 10 (16) | 10, 12, 11 (16) |
| | CLAVADDPM | 0, 0, 0 (9) | 5, 6, 5 (7) | 2, 2, 2 (16) | 2, 1, 2 (16) | 3, 3, 3 (16) | **0, 0, 0 (7)** | 3, 3, 5 (16) | **6, 6, 7 (16)** |
| | **BN** | **0, 0, 0 (9)** | 7, 7, 7 (7) | 2, 2, 2 (16) | 2, 2, 2 (16) | 3, 3, 3 (16) | **0, 0, 0 (7)** | **1, 1, 1 (16)** | 7, 7, 7 (16) |
| | CTGAN | 5, 7, 4 (9) | 7, 7, 7 (7) | 3, 3, 3 (16) | 3, 3, 3 (16) | 3, 5, 4 (16) | **0, 0, 0 (7)** | 7, 8, 7 (16) | 11, 13, 11 (16) |
| | DDPM | 6, 6, 6 (9) | 7, 7, 7 (7) | 4, 4, 3 (16) | 4, 4, 3 (16) | 4, 5, 4 (16) | 1, 2, 2 (7) | 9, 9, 7 (16) | 13, 13, 13 (16) |
| | NFLOW | 7, 8, 6 (9) | 7, 7, 6 (7) | 6, 6, 8 (16) | 6, 6, 7 (16) | 8, 8, 8 (16) | 1, 1, 1 (7) | 10, 11, 8 (16) | 14, 15, 13 (16) |
| | TVAE | 6, 6, 6 (9) | 7, 7, 7 (7) | 3, 3, 3 (16) | 3, 3, 3 (16) | 3, 3, 3 (16) | **0, 0, 0 (7)** | 8, 8, 8 (16) | 12, 12, 12 (16) |
| Walmart | SDV | 4, 4, 4 (4) | 8, 10, 8 (13) | 2, 2, 2 (17) | 2, 2, 2 (17) | 3, 3, 3 (17) | 1, 1, 1 (13) | 8, 7, 8 (17) | 17, 16, 17 (17) |
| | RCTGAN | 3, 3, 2 (4) | 11, 10, 9 (13) | 1, 1, 1 (17) | 1, 0, 1 (17) | 1, 1, 1 (17) | **0, 0, 0 (13)** | 10, 8, 9 (17) | 15, 16, 15 (17) |
| | REALTABF. | 3, 3, 3 (4) | 6, 9, 11 (13) | 4, 4, 4 (17) | 4, 4, 4 (17) | 4, 6, 7 (17) | 2, 2, 2 (13) | 7, 12, 11 (17) | 12, 12, 14 (17) |
| | MOSTLYAI | 4, 3, 3 (4) | 3, 3, 3 (13) | 3, 2, 2 (17) | 3, 2, 2 (17) | 3, 3, 3 (17) | **0, 0, 0 (13)** | 5, 5, 5 (17) | 9, 7, 7 (17) |
| | G-ACTGAN | 1, 1, 2 (4) | 11, 11, 12 (13) | 1, 1, 1 (17) | 1, 1, 1 (17) | 3, 3, 2 (17) | **0, 0, 0 (13)** | 6, 6, 10 (17) | 14, 14, 14 (17) |
| | G-LSTM | 2, 2, 1 (4) | **2, 2, 2 (13)** | 1, 1, 1 (17) | 1, 1, 1 (17) | 1, 1, 1 (17) | **0, 0, 0 (13)** | 4, 4, 3 (17) | **4, 4, 3 (17)** |
| | **CLAVADDPM** | **1, 1, 1 (4)** | 2, 4, 2 (13) | **0, 0, 0 (17)** | **0, 0, 0 (17)** | **0, 0, 0 (17)** | **0, 0, 0 (13)** | 1, 5, 2 (17) | 5, 4, 5 (17) |
| | BN | **1, 1, 1 (4)** | 4, 4, 4 (13) | **0, 0, 0 (17)** | **0, 0, 0 (17)** | 4, 5, 4 (17) | **0, 0, 0 (13)** | **2, 2, 2 (17)** | 10, 10, 10 (17) |
| | CTGAN | 3, 3, 1 (4) | 11, 12, 9 (13) | 1, 1, 1 (17) | 1, 1, 1 (17) | 2, 2, 1 (17) | **0, 0, 0 (13)** | 10, 10, 7 (17) | 14, 14, 13 (17) |
| | DDPM | **1, 1, 1 (4)** | 11, 11, 11 (13) | 8, 8, 8 (17) | 7, 7, 7 (17) | 8, 8, 8 (17) | 7, 7, 7 (13) | 9, 9, 9 (17) | 13, 13, 13 (17) |
| | NFLOW | 3, 2, 2 (4) | 8, 8, 8 (13) | 1, 1, 1 (17) | 1, 1, 1 (17) | 2, 2, 2 (17) | **0, 0, 0 (13)** | 8, 9, 9 (17) | 13, 13, 12 (17) |
| | TVAE | 2, 2, 2 (4) | 12, 12, 12 (13) | 1, 1, 1 (17) | 1, 1, 1 (17) | 2, 2, 2 (17) | **0, 0, 0 (13)** | 6, 6, 6 (17) | 14, 14, 14 (17) |
| Biodeg. | SDV | 3, 3, 3 (3) | 1, 1, 1 (3) | 2, 2, 2 (6) | 2, 2, 2 (6) | 2, 2, 2 (6) | 0, 0, 0 (3) | 3, 3, 3 (6) | 6, 6, 6 (6) |
| | **RCTGAN** | **1, 0, 1 (3)** | 3, 2, 2 (3) | **0, 0, 0 (6)** | **0, 0, 0 (6)** | **0, 0, 0 (6)** | 0, 0, 0 (3) | **1, 3, 1 (6)** | **3, 2, 4 (6)** |
| | MOSTLYAI | 2, 2, 2 (3) | 1, 1, 1 (3) | 2, 2, 2 (6) | 2, 2, 2 (6) | 2, 2, 2 (6) | 0, 0, 0 (3) | 3, 3, 2 (6) | **3, 3, 3 (6)** |
| | G-ACTGAN | 1, 1, 1 (3) | 3, 3, 3 (3) | **0, 0, 0 (6)** | **0, 0, 0 (6)** | **0, 0, 0 (6)** | 0, 0, 0 (3) | 3, 3, 3 (6) | 4, 4, 4 (6) |
| | G-LSTM | 3, 3, 3 (3) | **0, 0, 0 (3)** | 2, 2, 0 (6) | 2, 2, 0 (6) | 2, 2, 1 (6) | 0, 0, 0 (3) | 5, 5, 3 (6) | **3, 3, 3 (6)** |
| | BN | 2, 2, 2 (3) | 1, 1, 1 (3) | 2, 2, 2 (6) | 2, 2, 2 (6) | 2, 2, 2 (6) | 0, 0, 0 (3) | 2, 2, 2 (6) | **3, 3, 3 (6)** |
| | CTGAN | 3, 3, 3 (3) | 3, 3, 3 (3) | 2, 3, 3 (6) | 2, 3, 3 (6) | 3, 5, 3 (6) | 0, 0, 0 (3) | 5, 6, 5 (6) | 6, 6, 6 (6) |
| | DDPM | 2, 2, 2 (3) | 1, 1, 1 (3) | **0, 0, 0 (6)** | **0, 0, 0 (6)** | **0, 0, 0 (6)** | 0, 0, 0 (3) | 2, 2, 2 (6) | 5, 5, 5 (6) |
| | NFLOW | 3, 2, 3 (3) | 2, 2, 2 (3) | 2, 2, 2 (6) | 2, 2, 2 (6) | 3, 2, 3 (6) | 0, 0, 0 (3) | 4, 4, 4 (6) | 6, 5, 6 (6) |
| | TVAE | 3, 3, 3 (3) | 3, 3, 3 (3) | 2, 2, 2 (6) | 2, 2, 2 (6) | 2, 2, 2 (6) | 0, 0, 0 (3) | 5, 5, 5 (6) | 6, 6, 6 (6) |
| MovieLens | RCTGAN | 4, 4, 4 (7) | 3, 6, 6 (7) | 2, 2, 2 (14) | 2, 2, 2 (14) | 3, 3, 3 (14) | **0, 0, 0 (7)** | 8, 10, 9 (14) | 9, 11, 10 (14) |
| | MOSTLYAI | 5, 2, 3 (7) | 1, 4, 2 (7) | 3, 3, 3 (14) | 3, 3, 3 (14) | 3, 3, 3 (14) | 1, 1, 1 (7) | 6, 5, 4 (14) | 6, 8, 5 (14) |
| | G-ACTGAN | 4, 4, 4 (7) | 6, 6, 7 (7) | **0, 0, 0 (14)** | **0, 0, 0 (14)** | **0, 0, 0 (14)** | **0, 0, 0 (7)** | 9, 9, 9 (14) | 10, 10, 10 (14) |
| | **CLAVADDPM** | **2, 2, 2 (7)** | **0, 0, 0 (7)** | **0, 0, 0 (14)** | **0, 0, 0 (14)** | **0, 0, 0 (14)** | **0, 0, 0 (7)** | **3, 3, 2 (14)** | **2, 3, 2 (14)** |
| | DDPM | 3, 3, 3 (7) | 2, 2, 2 (7) | 5, 5, 5 (14) | 5, 5, 5 (14) | 5, 5, 5 (14) | 2, 2, 2 (7) | 5, 5, 5 (14) | 5, 5, 5 (14) |
| CORA | SDV | 2, 2, 2 (2) | - | 1, 1, 1 (2) | 1, 1, 1 (2) | 1, 1, 1 (2) | - | 2, 2, 2 (2) | 2, 2, 2 (2) |
| | **RCTGAN** | **0, 0, 0 (2)** | - | **0, 0, 0 (2)** | **0, 0, 0 (2)** | **0, 0, 0 (2)** | - | **0, 0, 0 (2)** | **0, 0, 0 (2)** |
| | G-ACTGAN | 1, 1, 1 (2) | - | 1, 1, 1 (2) | 1, 1, 1 (2) | 1, 1, 1 (2) | - | 1, 1, 1 (2) | 1, 1, 1 (2) |
| | G-LSTM | 2, 2, 2 (2) | - | 1, 1, 1 (2) | 1, 1, 1 (2) | 2, 1, 1 (2) | - | 2, 1, 2 (2) | 2, 1, 2 (2) |
| | BN | 1, 1, 1 (2) | - | 1, 1, 1 (2) | 1, 1, 1 (2) | 1, 1, 1 (2) | - | 1, 1, 1 (2) | 1, 1, 1 (2) |
| | CTGAN | 2, 2, 2 (2) | - | 2, 2, 2 (2) | 2, 2, 2 (2) | 2, 2, 2 (2) | - | 2, 2, 2 (2) | 2, 2, 2 (2) |
| | DDPM | 1, 1, 1 (2) | - | 1, 1, 1 (2) | 1, 1, 1 (2) | 1, 1, 1 (2) | - | 1, 1, 1 (2) | 1, 1, 1 (2) |
| | NFLOW | 2, 2, 2 (2) | - | 1, 1, 1 (2) | 1, 1, 1 (2) | 2, 2, 2 (2) | - | 2, 2, 2 (2) | 2, 2, 2 (2) |
| | TVAE | 2, 2, 2 (2) | - | 1, 1, 1 (2) | 1, 1, 1 (2) | 1, 1, 1 (2) | - | 2, 2, 2 (2) | 2, 2, 2 (2) |

Table 10: **Single-Table Results.** We report the number of times the method failed the fidelity test. There are three numbers for each combination, one for each replication. The number in parentheses is the total number of eligible tables for the corresponding metric. Note that REALTABFORMER does not support non-linear relational data (Biodegradability, CORA, MovieLens). CLAVADDPM is unable to model datasets with multiple foreign keys between pairs of tables (Biodegradability, CORA). SDV (timeout) and G-LSTM (missing tables) failed for the IMDB MovieLens dataset. MOSTLYAI (failed job) failed for the CORA dataset.

| Dataset | Method | Distance | | Detection | |
|---|---|---|---|---|---|
| | | MMD | PCD | LD | XGB |
| AirBnB | **SDV** | **0, 0, 0 (2)** | **0, 0, 0 (1)** | 2, 2, 2 (2) | 2, 2, 2 (2) |
| | **RCTGAN** | **0, 0, 0 (2)** | **0, 0, 0 (1)** | 2, 2, 2 (2) | 2, 2, 2 (2) |
| | REALTABF. | 2, 1, 1 (2) | 1, 1, 1 (1) | **2, 1, 1 (2)** | **2, 1, 1 (2)** |
| | **MOSTLYAI** | **0, 0, 0 (2)** | **0, 0, 0 (1)** | 2, 2, 2 (2) | 2, 2, 2 (2) |
| | G-ACTGAN | 1, 1, 1 (2) | 1, 1, 1 (1) | 2, 2, 2 (2) | 2, 2, 2 (2) |
| | G-LSTM | 0, 1, 0 (2) | 0, 0, 1 (1) | 2, 2, 2 (2) | 2, 2, 2 (2) |
| | CLAVADDPM | 1, 1, 1 (2) | 1, 1, 1 (1) | 2, 2, 2 (2) | 2, 2, 2 (2) |
| | **BN** | **0, 0, 0 (2)** | **0, 0, 0 (1)** | 2, 2, 2 (2) | 2, 2, 2 (2) |
| | CTGAN | 0, 1, 0 (2) | **0, 0, 0 (1)** | 2, 2, 2 (2) | 2, 2, 2 (2) |
| | DDPM | 0, 1, 0 (2) | 1, 1, 0 (1) | 2, 2, 2 (2) | 2, 2, 2 (2) |
| | NFLOW | 1, 1, 1 (2) | 1, 1, 1 (1) | 2, 2, 2 (2) | 2, 2, 2 (2) |
| | **TVAE** | **0, 0, 0 (2)** | **0, 0, 0 (1)** | 2, 2, 2 (2) | 2, 2, 2 (2) |
| Rossmann | SDV | 1, 1, 1 (2) | **0, 0, 0 (2)** | 2, 2, 2 (2) | 2, 2, 2 (2) |
| | RCTGAN | 1, 1, 0 (2) | **0, 0, 0 (2)** | 2, 2, 2 (2) | 2, 2, 2 (2) |
| | REALTABF. | 2, 2, 2 (2) | **0, 0, 0 (2)** | 2, 2, 2 (2) | 2, 2, 2 (2) |
| | MOSTLYAI | 1, 1, 1 (2) | **0, 0, 0 (2)** | 1, 2, 2 (2) | 2, 2, 2 (2) |
| | G-ACTGAN | **0, 0, 0 (2)** | **0, 0, 0 (2)** | 2, 2, 2 (2) | 2, 2, 2 (2) |
| | G-LSTM | 0, 1, 1 (2) | **0, 0, 0 (2)** | 2, 2, 2 (2) | **1, 2, 1 (2)** |
| | **CLAVADDPM** | **0, 0, 0 (2)** | **0, 0, 0 (2)** | 1, 0, 1 (2) | 2, 2, 2 (2) |
| | BN | **0, 0, 0 (2)** | **0, 0, 0 (2)** | 1, 1, 1 (2) | 2, 2, 2 (2) |
| | CTGAN | 1, 0, 0 (2) | **0, 0, 0 (2)** | 2, 2, 2 (2) | 2, 2, 2 (2) |
| | DDPM | 1, 1, 1 (2) | 0, 1, 1 (2) | 2, 2, 2 (2) | 2, 2, 2 (2) |
| | NFLOW | 1, 2, 1 (2) | **0, 0, 0 (2)** | 2, 2, 2 (2) | 2, 2, 2 (2) |
| | TVAE | **0, 0, 0 (2)** | **0, 0, 0 (2)** | 2, 2, 2 (2) | 2, 2, 2 (2) |
| Walmart | SDV | 2, 2, 2 (3) | 2, 2, 2 (2) | 3, 3, 3 (3) | 3, 3, 3 (3) |
| | RCTGAN | 2, 2, 1 (3) | 1, 1, 1 (2) | 3, 2, 2 (3) | 3, 3, 3 (3) |
| | REALTABF. | 2, 2, 2 (3) | 1, 1, 1 (2) | 2, 3, 2 (3) | 2, 2, 2 (3) |
| | MOSTLYAI | 2, 1, 1 (3) | 2, 2, 2 (2) | 3, 2, 2 (3) | 3, 3, 3 (3) |
| | G-ACTGAN | 1, 1, 1 (3) | 1, 1, 1 (2) | 2, 2, 2 (3) | 3, 3, 2 (3) |
| | **G-LSTM** | **0, 0, 0 (3)** | **0, 0, 0 (2)** | **1, 1, 1 (3)** | **1, 1, 1 (3)** |
| | CLAVADDPM | **0, 0, 0 (3)** | **0, 0, 0 (2)** | 1, 2, 1 (3) | 2, 2, 2 (3) |
| | BN | **0, 0, 0 (3)** | **0, 0, 0 (2)** | 1, 1, 1 (3) | 2, 2, 2 (3) |
| | CTGAN | 1, 1, 0 (3) | **0, 0, 0 (2)** | 3, 2, 2 (3) | 3, 3, 3 (3) |
| | DDPM | 1, 1, 1 (3) | 1, 1, 1 (2) | 2, 2, 2 (3) | 3, 3, 3 (3) |
| | NFLOW | 0, 1, 1 (3) | 1, 1, 1 (2) | 2, 2, 2 (3) | 2, 3, 3 (3) |
| | TVAE | **0, 0, 0 (3)** | **0, 0, 0 (2)** | **1, 1, 1 (3)** | 3, 3, 3 (3) |
| Biodeg. | SDV | **0, 0, 0 (1)** | **0, 0, 0 (1)** | 3, 3, 3 (4) | 4, 4, 4 (4) |
| | **RCTGAN** | **0, 0, 0 (1)** | 0, 1, 1 (1) | **0, 1, 2 (4)** | **1, 1, 2 (4)** |
| | MOSTLYAI | **0, 0, 0 (1)** | 1, 1, 1 (1) | 2, 2, 2 (4) | 3, 3, 3 (4) |
| | G-ACTGAN | **0, 0, 0 (1)** | **0, 0, 0 (1)** | 2, 2, 2 (4) | 2, 2, 2 (4) |
| | G-LSTM | **0, 0, 0 (1)** | **0, 0, 0 (1)** | 3, 3, 3 (4) | 3, 3, 3 (4) |
| | BN | **0, 0, 0 (1)** | **0, 0, 0 (1)** | 2, 2, 2 (4) | 3, 3, 3 (4) |
| | CTGAN | 1, 1, 0 (1) | **0, 0, 0 (1)** | 4, 4, 4 (4) | 4, 4, 4 (4) |
| | DDPM | **0, 0, 0 (1)** | **0, 0, 0 (1)** | 2, 2, 2 (4) | 3, 3, 3 (4) |
| | NFLOW | **0, 0, 0 (1)** | **0, 0, 0 (1)** | 3, 3, 3 (4) | 4, 3, 4 (4) |
| | TVAE | **0, 0, 0 (1)** | **0, 0, 0 (1)** | 4, 4, 4 (4) | 4, 4, 4 (4) |
| MovieLens | RCTGAN | **0, 0, 0 (5)** | 0, 0, 0 (2) | 3, 4, 4 (7) | 5, 5, 5 (7) |
| | MOSTLYAI | 1, 1, 1 (5) | 0, 0, 0 (2) | 5, 6, 3 (7) | 6, 6, 6 (7) |
| | G-ACTGAN | **0, 0, 0 (5)** | 0, 0, 0 (2) | 5, 5, 4 (7) | 6, 6, 6 (7) |
| | **CLAVADDPM** | **0, 0, 0 (5)** | 0, 0, 0 (2) | **3, 3, 2 (7)** | **2, 2, 2 (7)** |
| | DDPM | 2, 2, 2 (5) | 0, 0, 0 (2) | 4, 4, 4 (7) | 4, 4, 4 (7) |
| CORA | SDV | - | - | 2, 2, 2 (2) | 2, 2, 2 (2) |
| | **RCTGAN** | - | - | **0, 0, 0 (2)** | **0, 0, 0 (2)** |
| | G-ACTGAN | - | - | 1, 1, 1 (2) | 1, 1, 1 (2) |
| | G-LSTM | - | - | 2, 1, 2 (2) | 2, 1, 2 (2) |
| | BN | - | - | 1, 1, 1 (2) | 1, 1, 1 (2) |
| | CTGAN | - | - | 2, 2, 2 (2) | 2, 2, 2 (2) |
| | DDPM | - | - | 1, 1, 1 (2) | 1, 1, 1 (2) |
| | NFLOW | - | - | 2, 2, 2 (2) | 2, 2, 2 (2) |
| | TVAE | - | - | 2, 2, 2 (2) | 2, 2, 2 (2) |