# OpenReview forum: "Benchmarking the Fidelity and Utility of Synthetic Relational Data"
_ICLR.cc/2025/Conference — Submitted to ICLR 2025_

### Official Review · Reviewer_Ve6P · 2024-10-19

**Soundness:** 1
**Presentation:** 2
**Contribution:** 1
**Rating:** 3
**Confidence:** 5

**Summary:**

This paper addresses the challenge of relational data synthesis by introducing a general approach termed 'discriminative detection' to evaluate the fidelity of various relational data synthesis algorithms. The authors compare several recent solutions, including those utilizing Generative Adversarial Networks (GANs), and conduct an experimental study to assess the performance of these methods across different datasets.

**Strengths:**

The paper presents a good vision for establishing a benchmark that effectively evaluates both the fidelity and utility of synthesized relational data. Additionally, the initiative to define a generic metric is commendable, as it enhances the framework for assessing various synthesis methods.

**Weaknesses:**

The paper has several key weaknesses:

W1. Limited Identification of State-of-the-Art Methods: The authors only reference a few papers and overlook recent works exploring GANs and diffusion models for tabular data synthesis. For example, the VLDB Journal 2024 article below systematically analyzes the design space of tabular data synthesis using GANs, while studies like 'TableDiffusion' also utilize diffusion models. A comprehensive benchmark requires an up-to-date summary of SOTA methods.

- Tabular data synthesis with generative adversarial networks: design space and optimizations, VLDBJ 2024
- Diffusion Models for Tabular Data Imputation and Synthetic Data Generation (https://arxiv.org/pdf/2407.02549)

W2. Unclear Claims about Related Work: The statement regarding the lack of accessible APIs or source code for related studies is ambiguous. Notably, source code for significant works, such as the mentioned GAN paper and others that combine diffusion models, is publicly available. This oversight suggests that the authors may not have fully explored existing resources.

- https://github.com/ruc-datalab/Daisy for the above VLDBJ paper
- Tabsyn (https://github.com/amazon-science/tabsyn) that combines a diffusion model and a VAE,
- CoDi (https://github.com/ChaejeongLee/CoDi) also uses diffusion models
- and there are many more papers with code using GANs and diffusion models

W3. Insufficient Dataset Variety: A robust benchmark necessitates a wide array of datasets, yet this paper includes only five. In contrast, other studies, like the one on diffusion models, utilize ten datasets for evaluation, indicating that the current work lacks adequate data diversity.

- Diffusion Models for Tabular Data Imputation and Synthetic Data Generation (https://arxiv.org/pdf/2407.02549)

W4. Neglect of Data Type Considerations: The paper does not address the critical distinction between numerical (or continuous) and categorical (or discrete) data in tabular synthesis. Different data types require tailored generation techniques, which is a significant oversight in the discussion.

W5. Generalization of Fidelity Metrics: The fidelity evaluation should be context-dependent; thus, the authors must justify how a single metric can effectively encompass diverse applications before proposing a general metric.

W6. Lack of a Consistent Conclusion: The paper fails to provide a clear and consistent conclusion that can effectively guide researchers and practitioners in applying the findings.

**Questions:**

Q1: How many relational data synthesis methods in each category should be considered sufficient to establish a robust benchmark? (Refer to Weaknesses W1 and W2)

Q2: What is the minimum number of datasets required to create an effective benchmark? (Refer to Weaknesses W3 and W4)

Q3: How can real-world applications be used to motivate or justify the choice of a generic metric, and what makes this metric suitable for various real-world applications? (Refer to Weakness W5)

---

> ### Author Response · Authors · 2024-11-21
> **Addressing a Potential Misunderstanding**
>
> In the context of relational databases, the term **relation** is often used interchangeably with **table**, which may have contributed to a misunderstanding in the review. While there exists a version of [3] titled “Relational Data Synthesis using Generative Adversarial Networks: A Design Space Exploration,” both focus specifically on **tabular** data synthesis, not **relational** data synthesis. Our work addresses relational data synthesis and, as clarified in Section 2.1, clearly distinguishes it from tabular data synthesis. We kindly ask the reviewer to reevaluate their review and our manuscript in light of this distinction.
>
> **W1:** As stated, this work specifically focuses on methods for **relational** data synthesis—methods capable of synthesizing multiple tables connected by foreign key relationships—rather than **tabular** data synthesis. Consequently, we intentionally omitted several works that focus exclusively on tabular data synthesis, such as [1–9], as they are more appropriate for benchmarks evaluating tabular data synthesis [10].
>
> **W2:** We include all methods for **relational** data synthesis with available source code; see above response.
>
> **W3:** The mentioned study uses 10 tabular datasets, that is, **10 tables**. We include **6** relational datasets, for a total of **22 tables**. Comparing only single-table evaluation, our study includes more tables than the mentioned study. We focus on not only single table but multi table data, where evaluating the quality of individual tables is only one of the tasks in our evaluation.
>
> **W4:** We address the distinction between numerical (or continuous) and categorical (or discrete) data by including metrics which we specifically state are designed for numerical and categorical data, as well as add aggregations in our DDA metric specifically for numerical and categorical data respectively.
>
> **W5:** When it comes to fidelity, the context we are interested in is the specific aspect of the data we want to evaluate (e.g., single column fidelity or multi-table fidelity), and this is exactly what our approach enables, as clearly stated in Section 3.
>
> **W6:** One of our key findings is that current methods fail to generate synthetic relational data that is indistinguishable from the original. We updated the manuscript with further recommendations that focusing on the relational aspects of the data is a good direction for future work.
>
> **Q1:** We include all open source methods for relational data synthesis, as well as two commercial tools.
>
> **Q2:** We include a diverse collection of relational databases, including more datasets than are used in any of the related works.
>
> **Q3:**  Utility metrics are used to evaluate the performance of real-world applications of synthetic relational data, which we include in the paper. Fidelity metrics, on the other hand, are a general approach evaluating how well the synthetic data preserve the statistical properties of the original.
>
> [1] H. Zhang, J. Zhang, Z. Shen, B. Srinivasan, X. Qin, C. Faloutsos, H. Rangwala, G. Karypis, Mixed-type tabular data synthesis with score-based diffusion in latent space, in: The Twelfth International Conference on Learning Representations, 2024.
>
> [2] Shi, Juntong, et al. "TabDiff: a Unified Diffusion Model for Multi-Modal Tabular Data Generation." NeurIPS 2024 Third Table Representation Learning Workshop.
>
> [3] Liu, Tongyu, et al. "Tabular data synthesis with generative adversarial networks: design space and optimizations." The VLDB Journal 33.2 (2024): 255-280.
>
> [4] Lee, Chaejeong, Jayoung Kim, and Noseong Park. "Codi: Co-evolving contrastive diffusion models for mixed-type tabular synthesis." International Conference on Machine Learning. PMLR, 2023.
>
> [5] Kim, J., Lee, C., and Park, N. STasy: Score-based tabular data synthesis. In The Eleventh International Conference on Learning Representations, 2023
>
> [6] Zhao, Zilong, Robert Birke, and Lydia Chen. "Tabula: Harnessing language models for tabular data synthesis." arXiv preprint arXiv:2310.12746 (2023).
>
> [7] Gulati, Manbir, and Paul Roysdon. "TabMT: Generating tabular data with masked transformers." Advances in Neural Information Processing Systems 36 (2024).
>
> [8] Borisov, Vadim, et al. "Language models are realistic tabular data generators." arXiv preprint arXiv:2210.06280 (2022).
>
> [9] Liu, Tennison, et al. "GOGGLE: Generative modelling for tabular data by learning relational structure." The Eleventh International Conference on Learning Representations. 2023.
>
> [10] Qian, Zhaozhi, Rob Davis, and Mihaela van der Schaar. "Synthcity: a benchmark framework for diverse use cases of tabular synthetic data." Advances in Neural Information Processing Systems 36 (2024).

---

> > ### Comment · Reviewer_Ve6P · 2024-12-03
> >
> > 1. Single table synthesis is a special case of multi-table synthesis. Methods that can perform multi-table synthesis should also be able to perform single table synthesis. There are also recent studies from the database community that synthesize multiple tables or capture relationships from multiple papers, e.g., "ReStore - Neural Data Completion for Relational Databases".
> >
> > 2. "When it comes to fidelity, the context we are interested in is the specific aspect of the data we want to evaluate" -- this is not a justification from the user perspective. The justification should be why such a generic metric is what practitioners want.

---

> > > ### Author Response · Authors · 2024-12-04
> > >
> > > 1. We agree with the fact that single-table synthesis is a special case of multi-table synthesis, and this is why we thoroughly evaluated single-table fidelity for the relational methods and compared them to commonly used single-table baselines. We would like to stress again that, to the best of our knowledge, in our overview of relational generative methods, we included all available methods. The method you refer to deals with database **completion** and not **generation/synthesis**.
> > >
> > > 2. The metric is general in the sense that if a user is interested in single-column, single-table, or multi-table fidelity, they can use our metric in all these cases. As shown in the manuscript, such a metric can then clearly show how fidelity degrades as we transition from marginals to multiple tables. We believe that the generality and consequent practical applicability of the metric are what make it attractive for practitioners.

---

### Official Review · Reviewer_zEeQ · 2024-10-31

**Soundness:** 2
**Presentation:** 2
**Contribution:** 2
**Rating:** 3
**Confidence:** 4

**Summary:**

Experimental study that benchmarks various tools for generating synthetic relational data for a given source database. Uses standard approaches for single-table quality assessment (with straightforward but important modifications) and proposes a new approach for multi-table quality assessment. The main takeaway for me is that even single-table assessment fails with current data generation methods.

**Strengths:**

S1. Independent assessment and comparison of various synthetic data generation tools is a valuable contribution.

S2. The proposed "discriminative detection" for single-table assessment is a common, useful baseline approach.

**Weaknesses:**

W1. Technical novelty low. For single-table assessment, the proposed "discriminative detection" method is standard practice, as it amounts to training and evaluating a discriminator for real vs. synthetic examples. According to the authors, in the field of assessing synthetic relational data, prior work only used logistic regression as a discriminator. Although I find this hard to believe (e.g., some synthesis methods are GAN-based), I agree that that's took weak. For multi-table assessment, the proposed method is neither well-presented nor well-argued for. It computes/compares count and value statistics over joins, but it's not clear (i) why this is a good idea in the first place and (ii) which joins and which statistics should be used. For hierarchical data, it seems to modify the denormalization approach mainly by adding aggregation, but why/how does this actually lead to an improvement? Moreover, the approach seems to be basic and ignores the complex schemata of real relation data. As this is a benchmark paper, technical novelty is not a critical metric for assessment, but I include it here because the paper presents them as contributions.

W2. Insight low. I am not sure what I learned from this paper other than that even single-table assessment methods fail for the tested synthesizers. First, assuming that this wasn't known before already, the main reason must have been that a too weak model (logistic regression) was used as discriminator. That's a valuable insight, but it relevant to members of the subfield mainly, not the machine learning community as a whole (we know this). Second, given that single-table assessment already fails, doing multi-table assessment is not really helpful; it will also fail. In order to gauge the usefulness of the proposed aggregation method empirically, we want it to produce insights in cases where single-table assessment doesn't fail already. Third, the paper reports a large number of empirical results in tables and figures, but no further analysis, deeper discussion, or suggested next steps.

**Questions:**

None

---

> ### Author Response · Authors · 2024-11-21
>
> Thank you for your insightful review.
>
> **W1:** We agree that the technical novelty of “discriminative detection” itself might be low; it is in fact a classifier two sample test, as stated in Section 3.1. Our main contribution was packaging it as a general way of evaluating different aspects of relational data fidelity, be it marginal distributions, column pairs, individual tables, or multiple related tables.
> Furthermore, we pointed out that using only logistic regression as a discriminative model leads to a too lenient evaluation, on which we seem to agree. This is an important oversight, and, as we stated, only logistic regression was used in related work on relational data synthesis; however, this is also very common in the adjacent and more active field of tabular data synthesis, where papers published at this conference last year rely on logistic regression as a tool for evaluating fidelity (Zhang et al., Mixed-Type Tabular Data Synthesis with Score-based Diffusion in Latent Space, ICLR 2024).
>
> For multi-table fidelity evaluation, prior work uses two methods. The first, cardinality similarity, assesses only the preservation of foreign key cardinalities between table pairs, capturing a limited aspect of the data. The second, denormalization, joins table pairs into a single table, breaking the i.i.d. assumption on which most classifiers used in C2ST rely. This may result in scores deviating from the baseline 50% accuracy even for samples from the same distribution, undermining the premise of two-sample testing.
> Our metric combines elements of both approaches: it incorporates foreign key cardinalities and adds simple aggregations for numerical and categorical features from related tables. This way it tests cardinality as well as how well the relationships between columns in different tables are preserved. While these aggregations are indeed basic, they result in a metric that is already too difficult for all current methods to pass. As future methods mature past the current aggregations, different aggregations may be added to the metric as stated in Section 3.2.
>
> We updated the manuscript by justifying discriminative detection with aggregation as a propositionalization (of the C2ST for tabular data), a technique widely adopted in the fields of relational data mining and relational reasoning (Getoor et al. 2010).

---

> > ### Author Response · Authors · 2024-11-21
> >
> > **W2:** The insight of the paper and the relevance to the machine learning community are not in the specific generalization of the logistic detection metric but in the value of a rigorous benchmark for synthetic relational data (which is useful for tabular data as well) and the overview of the current state of the field, which is gaining a lot of traction in research and industry and, we believe, is only at its starting point. The paper provides a framework that gives insight into where exactly generative methods fail, highlights current weaknesses, and provides a way for future researchers to know where their method could be improved. Our evaluation also provides baselines for future work by making the comparison of new methods easier, as existing methods only compare themselves to the weak SDV method as a baseline. Our benchmark is provided as an open source Python package, which is designed to be easily extended in terms of new metrics and datasets, which we are sure will be added in the future by us and the community. As of right now, though, our collection of metrics and datasets is rigorous and sufficient to show the strengths and weaknesses of current approaches.
> >
> > We agree with your point that if single-table assessment already fails, multi-table assessment will also fail. We focused on two specific cases where single table fidelity did not fail; however, we admit that this was not communicated clearly. We updated our discussion of the results in Section 4.
> >  In our results section, we show:
> > - The SDV baseline, which is the only method current work compares their performance to, is too weak.
> > - Adding aggregations can highlight poorly generated relationships between tables, even when a single table fidelity test is passed. In figure 3 a) (SDV), we show an example where single table fidelity seems ok when using a subpar classifier (logistic regression), but adding aggregations shows the difference. When using a stronger classifier such as XGBoost, the single table fidelity test is passed by the G-LSTM method (3a, b); however, adding aggregations again shows that the relational fidelity is not satisfied, indicating the method fails to model relationships between tables.
> > - We continue the examination of the G-LSTM and ClavaDDPM methods in Figure 4, where we show that the classifier relied mainly on the aggregate columns to find the differences between the synthetic and original data as a single table fidelity test was passed.
> > - Lastly, we show how we can use our results to investigate which particular aspects of the data were poorly generated in figure 5a.
> >
> > We believe that future methods will pass single-table fidelity tests more often (as already indicated by the diffusion-based approach ClavaDDPM). Our metric DDA will then provide insight into the relational aspects, and we should not discourage research into relational data synthesis just because its single table fidelity is not yet perfect. We also added a paragraph discussing how, despite failing the fidelity test, the current methods achieve good utility on some datasets, indicating that they indeed have practical value.
> >
> > We updated the manuscript to highlight these findings more clearly, and based on these, we added recommendations for future work so that the message of the paper is clearer.
> >
> > Overall, we believe that synthetic relational data generation should not be overlooked. Relational databases hold a vast amount of the world’s data but often have limited utility due to holding sensitive information, and synthetic data is a promising solution to this.

---

> > > ### Comment · Reviewer_zEeQ · 2024-11-26
> > > **On W2**
> > >
> > > I disagree with your statements around Fig. 3. The argument taken in the response is that single-table classification fails because we use logistic regression, but as this paper shows and the ML community knows, logistic regression is too simple. As soon as XGB is used, the differences between single-table assessment and multi-table assessment become small.
> > >
> > > I agree with the authors on the other contributions (as I already wrote in my review), but I still feel that the paper is better suited to a more targeted venue and that there are problems around W1 and W2.

---

> > > > ### Author Response · Authors · 2024-12-02
> > > > **Response to Comments on W2**
> > > >
> > > > Our argument is that the use of logistic regression **during evaluation** has failed to show the shortcomings of single table generation, and therefore less emphasis was put on single table fidelity than perhaps necessary. It is important to state that currently single table fidelity (for both single and relational data) is estimated using only logistic regression as a C2ST and visualizing marginals and correlation plots (even in works accepted at high-end venues). However, this approach only evaluates linear relationships between columns.
> > > >
> > > > Our point was that in order to improve the fidelity of (tabular and relational) synthetic data, stronger classifiers should be used for evaluation. This is supported by your observation that using XGB highlights these weaknesses. The differences when using XGB are in fact small, as the tables are synthesized poorly; however, when individual tables are synthesized well, aggregations point out the current shortcomings in multi-table fidelity as stated in the manuscript.
> > > >
> > > > We had to condense much of these insights into the 10 available pages, and perhaps some of our explanations were lost in this process. We would nevertheless like to thank you for your insightful review; it has helped us improve the quality of the manuscript.

---

> > > > > ### Comment · Reviewer_zEeQ · 2024-12-02
> > > > > **Thanks for your response**
> > > > >
> > > > > I understand these points, but they do not directly address my concerns. I feel that this paper is not ready for the reasons mentioned, but I'd like to encourage you to continue this line of work.
> > > > >
> > > > > As for non-iid: I am not convinced by your argument. The key idea would be to consider samples from the normalized table, which then are iid. This may not be suitable for downstream tasks, but it may very well be suitable for assessing synthetic data.

---

> > > > > > ### Author Response · Authors · 2024-12-02
> > > > > > **Response to non-IID**
> > > > > >
> > > > > > In this context, as stated in the manuscript, denormalization is problematic from a perspective that the rows in the denormalized tables are not i.i.d. We believe this is enough justification that the accuracy of a discriminative classifier will not be 50%. For example, let's take a dataset with one parent and one child table connected in a one to many relationship. When denormalizing this relationship, the columns of the denormalized table will contain parent table columns and child table columns. Values of the parent table columns will be repeated for all of the child rows that are connected to the same parent row. This means that when splitting the denormalized dataset into train/test sets, the test set will, with high probability, include rows for which the parent table columns it has already seen in the train set. This means the model will be able to predict better than random. In fact, this is similar to our data copying experiment (Appendix D.2). There, however, the target labels for rows that have the same column values are different (an original row and then a copied synthetic row will have different target column values), which means that the classifier will see one row with one label in the train set and the same row with a different label in the test set.
> > > > > >
> > > > > > Regarding your comment of sampling the normalized table, in DDA, we are only considering samples from the normalized parent table, with added aggregation information. If you had a different way in mind we kindly ask you to provide further details.

---

> > > > > > > ### Comment · Reviewer_zEeQ · 2024-12-02
> > > > > > > **non-IID ctd.**
> > > > > > >
> > > > > > > Thanks & now understood!
> > > > > > >
> > > > > > > My main comment on this point would be what's in the original review: "For hierarchical data, it seems to modify the denormalization approach mainly by adding aggregation, but why/how does this actually lead to an improvement? Moreover, the approach seems to be basic and ignores the complex schemata of real relation data."
> > > > > > >
> > > > > > > My current understanding is that the main purpose of the aggregation approach is to become "value-agnostic".
> > > > > > >
> > > > > > > Be that as it may, I do not have specific alternative approaches in mind, but I do feel that a paper on multi-table fidelity assessment should explore such approaches. Next to simply using the denormalization approach, one simple baseline would be to build train/test splits with disjoint foreign keys. This will probably lead to over-fragmentation in practice (=won't work well), esp. when there are multiple foreign keys, but it is a simple alternative and one can do it for subsets of tables. Another option may be to permute values (akin to STUNT, NeurIPS22) or to replace/hide actual values.
> > > > > > >
> > > > > > > Again, I do feel that this line of work is valuable and deserves to be explored further.

---

> > > > > > > > ### Author Response · Authors · 2024-12-04
> > > > > > > >
> > > > > > > > Thank you for recognizing the value of our work. We regret that our presentation of discriminative detection detracted from our other contributions, as this is first and foremost a benchmark paper for relational generative methods.
> > > > > > > >
> > > > > > > > In this context, our aim was to create an exhaustive overview of related work in both generation and evaluation and combine them in a benchmark. We also wanted to clearly define fidelity in the context of the indistinguishability of synthetic and original data, as mentioned above. In our review, we found several flaws (e.g., using only denormalization for multi-table fidelity), which we point out and address in our manuscript. As there was a clear shortcoming in multi-table fidelity evaluation, we then proposed a metric that builds upon standard metrics for single table fidelity and a commonly used approach in statistical relational learning (propositionalization).
> > > > > > > >
> > > > > > > > As our interest was in establishing the quality of the methods for relational data generation, we were not particularly concerned with comparing evaluation metrics. Our proposed metric is intentionally 1) simple, 2) based on well-established approaches, and 3) practically useful, as demonstrated in our results. As the above-mentioned shortcomings of denormalization clearly disqualify it in the context of indistinguishability, we believe such a comparison was not necessary.
> > > > > > > >
> > > > > > > > Aggregation essentially achieves exactly what you propose: the train-test splits are foreign-key disjoint (without over-fragmentation) as they are performed on a normalized table. Aggregation is also computed for all parent tables, so it provides a thorough overview of the generated data. The information from child tables must be combined in some form in order to use classical ML approaches. An alternative would be to use graph-based approaches, the application of which is not clear for detection-based fidelity.
> > > > > > > >
> > > > > > > > Once again, thank you for your insightful review and encouragement, and we hope this clarifies the intent and contributions of our work.

---

> > ### Comment · Reviewer_zEeQ · 2024-11-26
> > **On W1**
> >
> > Thanks for the update! The new manuscript didn't mark changes, so I'll just go with what you write.
> >
> > My main concern w.r.t. discriminative detection is that the paper positions it as a contribution, whereas this is common in the wider ML community.
> >
> > My main concern with multi-table fidelity is that the approach is not sufficiently justified, both w.r.t. its goals and its limitations. The "denormalization baseline" is also missing in the experimental study, ideally directly with DD. (In fact, I am not convinced that the denormalization approach is problematic for the purpose of fidelty estimation.)

---

> ### Author Response · Authors · 2024-12-02
> **Response to Comments on W1**
>
> Just to clarify, our contribution was DD with aggregation; however, we first framed the metric as DD to allow different classifiers and transformations, which is new in the synthetic data community and results in a general metric for fidelity.
>
> Our aim was to frame fidelity evaluation in the context of the indistinguishability of synthetic and relational data. For this reason some form of decision function is necessary to observe if the synthetic and original data are statistically indistinguishable. We define these for all of the available metrics. When using a discriminative model to distinguish between two samples, a natural decision function is the deviation of the classifier from random performance.
>
> As stated in the manuscript, denormalization is problematic because it breaks the IID assumption, and with that, the guarantee that the classifier can achieve >50% accuracy if and only if we can discriminate between synthetic and real data. Detecting poor fidelity not because of poor data generation by the method but because of the characteristics of the evaluation procedure is not just problematic; it makes the procedure useless in the context of indistinguishability.

---

### Official Review · Reviewer_4F3U · 2024-11-03

**Soundness:** 3
**Presentation:** 3
**Contribution:** 3
**Rating:** 6
**Confidence:** 4

**Summary:**

As pointed out by the authors, the use of synthetic relational data is
attractive as it can help to preserve the privacy of the original
data, and it can help to alleviate the scarcity of data. Of course, to
help with these problems and be useful, the synthetic data has to
preserve some properties of the original data. This is an old problem
that has been studied in the database area; in particular, in this
context, the type of properties one would like to preserve in the
synthetic data are related to the complexity of evaluating certain
classes of queries in the real and synthetic scenarios. However, what
is new in the scenario presented in this paper is that an ML model is
going to be trained on the synthetic data to later make predictions on
the real data. Hence, the synthetic data has to preserve the structure
of the real data that allows the ML model to make accurate
predictions on the real data.

The main aspects used to measure the quality of synthetic relational
data are fidelity and utility. The former refers to how close the
synthetic data is to the real data, while the latter refers to how
well an ML model would perform on a prediction task over the real data
if the real data is replaced by the synthetic data. In this paper, the
authors propose a synthetic relational data benchmark, which combines
known techniques to measure fidelity and utility with a new general
approach to measure fidelity for relational data. Moreover, the
authors use this approach to compare known methods for synthesizing
relational data.

**Strengths:**

S1) The authors present a fairly general framework to compare methods
for synthesizing relational data.

S2) The paper presents an experimental evaluation where the proposed
framework is used to compare known methods for synthesizing relational
data. This evaluation provides useful information about these methods.

**Weaknesses:**

W1) The general approach to measure fidelity with discriminative
detection can be understood. However, the notion of relational data
used in the paper is not properly formalized.

**Questions:**

The notion of relational data used in the paper is not properly formalized.

- The authors mention that the datasets used in the paper are
  organized based on the structure of their relational schema. For
  example, they mention that AirBnB uses only linear relationships,
  which means one parent and one child table. But they do not indicate
  how the parent-child relationship is defined between tables. Is this
  relationship defined considering the foreign keys in the tables?

- A foreign key for the table T_i is defined p_{T_j}, T_i ~ T_j. I
  assumed this foreign key is defined with respect to the primary key
  of the table T_j, since this foreign key is defined for T_i. But
  which attributes of T_i participate in this dependency? The notation
  has to include this information.

- A row of a table is defined as a set of values. Since a set is not
  ordered, how do you know what is the value of an attribute of the
  table in this row?

- The authors indicate that if {a_1^{T_i}, ..., a_l^{T_i}} are the
  attributes of a table T_i, then a row of this table is a set of
  values {v_{p_{T_i}}, v_{k_1}, ..., v_{k_o}, v_{a_1^{T_i}}, ...,
  v_{a_l^{T_i}}}. I assume that v_{a_j^{T_i}} is the value of
  attribute a_j^{T_i}. But then I do not understand what the other
  values are, as they do not correspond with the attributes of the
  table.

---

> ### Author Response · Authors · 2024-11-21
>
> **W1:** We would like to thank you for pointing out this oversight, which we clarified in the manuscript.
>
> **Q1:** The relationship is indeed defined based on the foreign keys in the tables; we made this distinction clearer.
>
> **Q2:** Our original definition was vague in which attributes are related to foreign keys; we clarified this by incorporating elements from the accepted definition from Fey et al. (2024).
>
> **Q3:** We agree that defining the row of a table as a set is problematic. Our intention was to highlight the permutation invariance between the columns of a table. However, a more suitable definition would be a set of key value pairs {(k_i, v_i,j)}, where k_i is the name of the attribute and v_i,j is the value in the jth row.
>
> **Q4:** Those values are intended to hold the information of the foreign and primary keys. However, we did not state this explicitly, resulting in issues addressed in Q2.
>
>
> As stated, we refined our definition based on the accepted definition from “Fey et. al., Relational Deep Learning: Graph Representation Learning on Relational Databases, 2023, ICML 2024.”.

---

### Official Review · Reviewer_qXxX · 2024-11-06

**Soundness:** 2
**Presentation:** 2
**Contribution:** 1
**Rating:** 3
**Confidence:** 4

**Summary:**

This paper proposes a benchmark for evaluating synthetic data generation techniques specifically designed for relational data. Notably, it introduces a method to assess synthetic data generation based on fidelity metrics and reports benchmark evaluation results on existing relational data generation techniques.

**Strengths:**

Establishing benchmarks for synthetic relational data generation is an interesting contribution, particularly with a focus on fidelity and utility as critical evaluation metrics.

**Weaknesses:**

- W1: While the focus on benchmarking is indeed valuable, the evaluation experiments on the proposed benchmark and existing techniques appear insufficient. Specifically, in Table 1, the comparison between statistical cardinality and the proposed approach is discussed only briefly in Section 4.3, with only around four lines of explanation, leaving the analysis of the proposed method’s advantages underdeveloped. I suggest the authors redesign the whole experiment section to have a new question, such as "Q1: Is the proposed benchmark more effective than SOTAs?".

- W2: Although the primary purpose of the paper, as stated in Chapter 1, is **privacy protection**, it does not assess some major approaches in this field, such as relational data generation using differential privacy. I would suggest the authors either 1) to add several existing data generation techniques based on differential privacy, or 2) to explain why you exclude these techniques in spite of claiming privacy protection is the benefit of synthetic data generation.
Some SOTAs based on differential privacy are as follows:
  - J. Yang, P. Wu, G. Cong, T. Zhang, and X. He. “SAM: Database generation from query workloads with supervised autoregressive models.” In SIGMOD, 2022.
  - K. Cai, X. Xiao, and G. Cormode. “Privlava: Synthesizing relational data with foreign keys under differential privacy.” SIGMOD, 1(2), 2023.



- W3: The paper places a heavy emphasis on fidelity, with only limited evaluation of utility, creating an imbalance. Since the tile and abstract claim that fidelity and utility have the same importance in benchmarking synthetic data generation, the experiment of benchmarking should also have similar importance over fidelity and utility. For instance, granularity (single-column, single-table, multi-table) is extensively examined with regard to fidelity, taking up considerable space in the experiments, while utility is assessed only for single-table input, with a relatively small space devoted to these experiments. I would suggest adding more experiments for utility evaluation: 1) using regression/classification for estimating different attribute values, and/or 2) joining different numbers of tables and using them as input to train the models.

- W4: The proposed method in Section 3 appears relatively simple, so its novelty is not clear. In detail, Algorithm 1 simply trains the model to discriminate the original data and (derived) synthetic data using the Binominal test, which is just a standard technique. Also, the extension to multi-table in Section 3.2 is quite simple, in that it utilizes simple aggregation from child tables.
- W5: Issues with readability and consistency:
  - Several graphs in the experimental section lack descriptions on the X and Y axes.
  - In Figure 2, the description “Most methods model the parent table (store) better since the tests find more differences for the child table (historical)” applies to (b) but not to (a).
  - Inconsistencies exist between Section 4.5 (first and third most important features) and Figure 5 (1st and 4th most important features).

**Questions:**

- Q-error is a standard measure for evaluating record count accuracy in query results, but this paper does not adopt Q-error. What is the reason for this?
- Although the paper asserts the importance of machine learning utility, machine learning models typically only handle single-table structured data as input. Conversely, the paper emphasizes the importance of multi-table data generation, creating a potential inconsistency in its argument. How does the paper propose to input multi-table data into machine learning models? Additionally, how is the input data prepared from multiple tables for Section 4.6?
- The phrase “generating child rows conditionally on parent rows” implies a propagation of information to the parent side. However, the paper associates this with “propagating errors down the hierarchy,” referring to propagation to the child side, which seems inconsistent.

---

> ### Author Response · Authors · 2024-11-21
>
> **W1:** We address current SOTA evaluation tools and their weaknesses in the related work section. We introduce the **first** benchmark that consolidates best practices from SOTA methods and includes improved metrics—extending logistic detection to an arbitrary classifier, incorporating relational information, and adding explainability methods to detection metrics to facilitate model debugging. We discuss the advantages of the method over other metrics in Section 3.3. However, our intent was to evaluate the performance of the generative methods more so than comparing existing metrics. Given our descriptions of both metrics, we believe that the results are expected, and as such, our brief discussion in Section 4.3 is sufficient.
>
> **W2:** The primary objective of this paper is to construct a procedure for evaluating the quality of generative methods for synthetic relational data. In principle, DP methods like PrivLava could be included in our comparison, but DP methods deliberately sacrifice fidelity to achieve DP. In the case of PrivLava, by injecting Gaussian noise. As the authors of PrivLava state in the conclusion, with fewer tuples, we need more noise to ensure DP (and less noise means there are more tuples available). This suggests a clear way of distinguishing between the original dataset and a synthetic dataset with DP. So we find it unlikely that the fidelity performance of DP methods would be state-of-the-art. DP literature also indirectly acknowledges that it is not competing in fidelity, as most papers do not include any fidelity evaluation and there is very little comparison with state-of-the-art synthetic data generation methods. We added PrivLava to the generative methods survey in Appendix A; however, SAM is specifically intended for generating databases from query workloads, which are not defined for our datasets, and for this reason we do not consider it in this work (see our response to Q1).
>
> **W3:** We placed particular emphasis on multi-table fidelity and utility because we are evaluating synthetic relational data. In fact, we are the first to evaluate multi-table utility, as related work uses only single-table utility. We could include additional results for single-table utility in the appendix if the reviewer believes they will significantly contribute to the merit of this work. To address your suggestions: 1) The datasets we include in this benchmark have specifically defined tasks, which is why we opted to use those in our utility experiments as is standard practice in tabular utility evaluation. 2) This is how we implemented utility evaluation. As we are dealing with relational data, our focus was on multi-table utility. We constructed pipelines that join related tables, which is typical in a data science workload when handling multiple tables. The pipelines are described in detail in Appendix C.3.
>
> **W4:** We agree that the method is simple; however, this does not disqualify it as a good metric. The C2ST is a widely adopted way of statistically comparing two samples of structured data; however, as we clearly stated, current work is using an extremely forgiving version of this algorithm for evaluation by relying on logistic regression as the underlying classifier (and often not using a binomial test but simply looking at the AUC value). This includes works accepted at high-end venues, such as ClavaDDPM. We build on this not only by adding a stronger classifier but also by generalizing the metric to different aspects of relational data and equipping it with interpretability approaches to facilitate insights into the shortcomings of generative methods. The extension to relational data is also simple; however, as we state in Section 3.2, different aggregations may be added, but they are currently not necessary, as current methods already fail when using 3 simple aggregations. We added a justification of the method with aggregation as a propositionalization approach to the C2ST on relational data.
>
> **W5:**
> - Could you please point out the specific examples of graphs without X and Y axis labels?
> - We made the description of Figure 2 more accurately reflect the results presented.
> - Thank you for pointing out the inconsistency between Section 4.5 and Figure 5. We fixed it.

---

> > ### Author Response · Authors · 2024-11-21
> >
> > **Q1:** Q-error is used to measure the error between a predicted cardinality and the true cardinality of a database query. As such, it assumes the existence of a query, which we do not have in our setting. We have two relational databases, one is synthetic and the other is not, and we are trying to answer if they were generated according to the same process. That is, we are trying to answer if the synthetic database is (in terms of its statistical properties) indistinguishable from real data.
> > Even if we came up with a query, we don't have the true cardinality. And the two databases need not be the same, and the cardinalities of the same query run on the two different databases need not be the same. We are concerned with differences in distribution, and with a single query, we only get two points. It would be possible to run multiple queries (for example, one for each record) and then compare the distributions of cardinalities. But this is now close to what we propose and far from Q-error. However, our benchmark is designed to be easily extensible, allowing for the future addition of Q-error as a utility metric along with relevant datasets with provided query sets.
> >
> > **Q2:** As stated in our response to W3, for utility we construct feature engineering pipelines (as is common practice for handling relational data for machine learning) in which we combine data from multiple tables based on foreign key relationships into one table. As stated in Section 4.6, the pipelines are described in detail in Appendix C.3. We also added a direction for future work to the manuscript by utilizing graph representation learning to evaluate multi-table fidelity.
> >
> > **Q3:** We do not believe that “generating child rows conditionally on parent rows” implies a propagation of information to the parent side; in fact, the opposite; the information is propagated towards the child. As most methods sample and model the data in an iterative fashion, starting with the parent and then generating child tables, the errors in fact propagate towards the child. For example, inconsistencies in sampling parent values will have an effect when sampling the child values, as these will be based on the poorly sampled parent values.

---

### Author Response · Authors · 2024-11-21

We would like to thank the reviewers for their time and quality of reviews. We would like to outline a few general points that are important for all reviews.

Synthetic relational data generation is a new field without well-established methods for quality evaluation. This includes evaluation metrics as well as which datasets to include in the evaluation. Additionally, the existing generative methods do not thoroughly compare themselves with each other. The main contribution of this paper is the benchmark, which addresses the above shortcomings. We conducted a comprehensive overview of metrics and datasets used in related work, outlined the strengths and, most importantly, the weaknesses in the evaluation procedures, and resolved them through our benchmark. The most notable weaknesses are:
- No comprehensive comparison to establish the merit of individual methods
- Lenient evaluation resulting from using logistic regression as a base classifier in C2ST.
- Reporting metric values without accounting for the variability of the original data.
- Shortcomings and a lack of metrics for evaluating the relational aspects of the data result in current approaches underperforming in this key aspect of  synthetic **relational** data.

We resolve these by:
- Gathering SOTA evaluation metrics for single-column, single-table, and relational data into an open-source benchmark and gathering all generative methods for relational data and thoroughly evaluating them.
- Addressing the weaknesses of current detection-based metrics.
- Formalizing the notion of distinguishability for distance, statistical, and detection-based metrics.
- Proposing a general detection-based approach suited to relational data, which we use to show which aspects of relational data were currently overlooked.

Our work creates an exhaustive overview of the field through empirical evaluation of all open source methods as well as two leading commercial tools on a diverse collection of relational databases. As such, it paves the way for future work by providing researchers with a baseline to measure against, defining how to perform the evaluation, and providing a metric (discriminative detection), which can help researchers understand what aspect of the data their method fails to address.

We submitted a revised version of the manuscript in which we:
- Extended our interpretation of the results refining the key insight of the paper.
- Added a synopsis of the results to the introduction and based on the results directions for future work to the conclusion.
- Clarified the definition of a relational database.
- Added a theoretical justification for using aggregation.

---

### Meta-Review · Area_Chair_Ladq · 2024-12-20

**Metareview:**

While the paper presents a general framework to compare methods for synthesizing relational data, there are also concerns on the novelty/contributions, technical details, related work, experiments, conclusion, and presentation. During the discussion period, the reviewers largely maintained their concerns. Overall, I think the paper can be improved and should go through another round of reviewing.

**Additional Comments On Reviewer Discussion:**

- Novelty/contributions: reviewer zEeQ has remaining concerns.
- Technical details: reviewer zEeQ has remaining concerns.
- Related work: reviewer Ve6P has remaining concerns.
- Experiments: reviewer Ve6P has remaining concerns.

---

### Decision · Program_Chairs · 2025-01-22

Reject